# LONG EXPRESSIVE MEMORY FOR SEQUENCE MODELING

**T. Konstantin Rusch**
ETH Zürich
trusch@ethz.ch

**Siddhartha Mishra**
ETH Zürich
smishra@ethz.ch

**N. Benjamin Erichson**
University of Pittsburgh
erichson@pitt.edu

**Michael W. Mahoney**
ICSI and UC Berkeley
mmahoney@stat.berkeley.edu

## ABSTRACT

We propose a novel method called *Long Expressive Memory* (LEM) for learning long-term sequential dependencies. LEM is gradient-based, it can efficiently process sequential tasks with very long-term dependencies, and it is sufficiently expressive to be able to learn complicated input-output maps. To derive LEM, we consider a system of *multiscale ordinary differential equations*, as well as a *suitable time-discretization* of this system. For LEM, we derive rigorous bounds to show the mitigation of the exploding and vanishing gradients problem, a well-known challenge for gradient-based recurrent sequential learning methods. We also prove that LEM can approximate a large class of dynamical systems to high accuracy. Our empirical results, ranging from image and time-series classification through dynamical systems prediction to keyword spotting and language modeling, demonstrate that LEM outperforms state-of-the-art recurrent neural networks, gated recurrent units, and long short-term memory models.

## 1 INTRODUCTION

Learning tasks with sequential data as inputs (and possibly outputs) arise in a wide variety of contexts, including computer vision, text and speech recognition, natural language processing, and time series analysis in the sciences and engineering. While recurrent gradient-based models have been successfully used in processing sequential data sets, it is well-known that training these models to process (very) long sequential inputs is extremely challenging on account of the so-called *exploding and vanishing gradients problem* (Pascanu et al., 2013). This arises as calculating hidden state gradients entails the computation of an iterative product of gradients over a large number of steps. Consequently, this (long) product can easily grow or decay exponentially in the number of recurrent interactions.

Mitigation of the exploding and vanishing gradients problem has received considerable attention in the literature. A classical approach, used in Long Short-Term Memory (LSTM) (Hochreiter & Schmidhuber, 1997) and Gated Recurrent Units (GRUs) (Cho et al., 2014), relies on *gating mechanisms* and leverages the resulting additive structure to ensure that gradients do not vanish. However, gradients might still explode, and learning very long-term dependencies remains a challenge for these architectures (Li et al., 2018). An alternative approach imposes constraints on the structure of the hidden weight matrices of the underlying recurrent neural networks (RNNs), for instance by requiring these matrices to be unitary or orthogonal (Henaff et al., 2016; Arjovsky et al., 2016; Wisdom et al., 2016; Kerg et al., 2019). However, constraining the structure of these matrices might lead to significantly reduced expressivity, i.e., the ability of the model to learn complicated input-output maps. Yet another approach relies on enforcing the hidden weights to lie within pre-specified bounds, leading to control on gradient norms. Examples include Li et al. (2018), based on *independent neurons* in each layer, and Rusch & Mishra (2021a), based on a network of coupled oscillators. Imposing such restrictions on weights might be difficult to enforce, and weight clipping could reduce expressivity significantly.

This brief survey highlights the challenge of *designing recurrent gradient-based methods for sequence modeling which can mitigate the exploding and vanishing gradients problem, while at the same time being sufficiently expressive and possessing the ability to learn complicated input-output maps efficiently.* We seek to address this challenge by proposing a novel gradient-based method.

The starting point for our method is the observation that realistic sequential data sets often contain information arranged according to multiple (time, length, etc., depending on the data and task) scales. Indeed, if there were only one or two scales over which information correlated, then a simple model with a parameter chosen to correspond to that scale (or, e.g., scale difference) should be able to model the data well. Thus, it is reasonable to expect that a *multiscale model* should be considered to process efficiently such *multiscale data*. To this end, we propose a novel gradient-based architecture, *Long Expressive Memory* (LEM), that is based on a suitable time-discretization of a set of multiscale ordinary differential equations (ODEs). For this novel gradient-based method (proposed in Section 2):

- we derive bounds on the hidden state gradients to prove that LEM mitigates the exploding and vanishing gradients problem (Section 4);

- we rigorously prove that LEM can approximate a very large class of (multiscale) dynamical systems to arbitrary accuracy (Section 4); and

- we provide an extensive empirical evaluation of LEM on a wide variey of data sets, including image and sequence classification, dynamical systems prediction, keyword spotting, and language modeling, thereby demonstrating that LEM outperforms or is comparable to state-of-the-art RNNs, GRUs and LSTMs in each task (Section 5).

We also discuss a small portion of the large body of related work (Section 3), and we provide a brief discussion of our results in a broader context (Section 6). Much of the technical portion of our work is deferred to Supplementary Materials.

## 2 LONG EXPRESSIVE MEMORY

We start with the simplest example of a system of *two-scale ODEs*,

$$\frac{d\mathbf{y}}{dt} = \tau_y \left( \sigma \left( \mathbf{W}_y \mathbf{z} + \mathbf{V}_y \mathbf{u} + \mathbf{b}_y \right) - \mathbf{y} \right), \quad \frac{d\mathbf{z}}{dt} = \tau_z \left( \sigma \left( \mathbf{W}_z \mathbf{y} + \mathbf{V}_z \mathbf{u} + \mathbf{b}_z \right) - \mathbf{z} \right). \tag{1}$$

Here, $t \in [0, T]$ is the continuous time, $0 < \tau_y \le \tau_z \le 1$ are the two time scales, $\mathbf{y}(t) \in \mathbb{R}^{d_y}, \mathbf{z}(t) \in \mathbb{R}^{d_z}$ are the vectors of *slow* and *fast* variables and $\mathbf{u} = \mathbf{u}(t) \in \mathbb{R}^m$ is the *input signal*. For simplicity, we set $d_y = d_z = d$. The dynamic interactions between the neurons are modulated by weight matrices $\mathbf{W}_{y,z}, \mathbf{V}_{y,z}$, bias vectors $\mathbf{b}_{y,z}$ and a *nonlinear* tanh activation function $\sigma(u) = \tanh(u)$. Note that $\odot$ refers to the componentwise product of vectors.

However, two scales (one fast and one slow), may not suffice in representing a large number of scales that could be present in realistic sequential data sets. Hence, we need to generalize (1) to a *multiscale* version. One such generalization is provided by the following set of ODEs,

$$\begin{aligned}
\frac{d\mathbf{y}}{dt} &= \hat{\sigma} \left( \mathbf{W}_2 \mathbf{y} + \mathbf{V}_2 \mathbf{u} + \mathbf{b}_2 \right) \odot \left( \sigma \left( \mathbf{W}_y \mathbf{z} + \mathbf{V}_y \mathbf{u} + \mathbf{b}_y \right) - \mathbf{y} \right), \\
\frac{d\mathbf{z}}{dt} &= \hat{\sigma} \left( \mathbf{W}_1 \mathbf{y} + \mathbf{V}_1 \mathbf{u} + \mathbf{b}_1 \right) \odot \left( \sigma \left( \mathbf{W}_z \mathbf{y} + \mathbf{V}_z \mathbf{u} + \mathbf{b}_z \right) - \mathbf{z} \right).
\end{aligned} \tag{2}$$

In addition to previously defined quantities, we need additional weight matrices $\mathbf{W}_{1,2}, \mathbf{V}_{1,2}$, bias vectors $\mathbf{b}_{y,z}$ and sigmoid activation function $\hat{\sigma}(u) = 0.5(1 + \tanh(u/2))$. As $\hat{\sigma}$ is monotone, we can set $\mathbf{W}_{1,2} = \mathbf{V}_{1,2} \equiv 0$ and $(\mathbf{b}_1)_j = b_y, (\mathbf{b}_2)_j = b_z$, for all $1 \le j \le d$, with $\hat{\sigma}(b_{y,z}) = \tau_{y,z}$ to observe that the two-scale system (1) is a special case of (2). One can readily generalize this construction to obtain many different scales in (2). Thus, we can interpret $(\boldsymbol{\tau}_z(\mathbf{y}, t), \boldsymbol{\tau}_y(\mathbf{y}, t)) = (\hat{\sigma} \left( \mathbf{W}_1 \mathbf{y} + \mathbf{V}_1 \mathbf{u} + \mathbf{b}_1 \right), \hat{\sigma} \left( \mathbf{W}_2 \mathbf{y} + \mathbf{V}_2 \mathbf{u} + \mathbf{b}_2 \right))$ in (2) as input and state dependent gating functions, which endow ODE (2) with *multiple time scales*. These scales can be learned adaptively (with respect to states) and dynamically (in time). Moreover, it turns out that the multiscale ODE system (2) is of the same general form (see **SM**§C) as the well-known Hodgkin-Huxley equations modeling the dynamics of the action potential for voltage-gated ion-channels in biological neurons (Hodgkin & Huxley, 1952).

Next, we propose a time-discretization of the multiscale ODE system (2), providing a circuit to our sequential model architecture. As is common with numerical discretizations of ODEs, doing so properly is important to preserve desirable properties. To this end, we fix $\Delta t > 0$, and we discretize (2) with the following implicit-explicit (IMEX) time-stepping scheme to arrive at LEM, written in compact form as,

$$
\begin{aligned}
\boldsymbol{\Delta t}_n &= \Delta t \hat{\sigma}(\mathbf{W}_1 \mathbf{y}_{n-1} + \mathbf{V}_1 \mathbf{u}_n + \mathbf{b}_1), \\
\overline{\boldsymbol{\Delta t}}_n &= \Delta t \hat{\sigma}(\mathbf{W}_2 \mathbf{y}_{n-1} + \mathbf{V}_2 \mathbf{u}_n + \mathbf{b}_2), \\
\mathbf{z}_n &= (1 - \boldsymbol{\Delta t}_n) \odot \mathbf{z}_{n-1} + \boldsymbol{\Delta t}_n \odot \sigma(\mathbf{W}_z \mathbf{y}_{n-1} + \mathbf{V}_z \mathbf{u}_n + \mathbf{b}_z), \\
\mathbf{y}_n &= (1 - \overline{\boldsymbol{\Delta t}}_n) \odot \mathbf{y}_{n-1} + \overline{\boldsymbol{\Delta t}}_n \odot \sigma(\mathbf{W}_y \mathbf{z}_n + \mathbf{V}_y \mathbf{u}_n + \mathbf{b}_y).
\end{aligned}
\tag{3}
$$

For steps $1 \le n \le N$, the hidden states in LEM (3) are $\mathbf{y}_n, \mathbf{z}_n \in \mathbb{R}^d$, with input state $\mathbf{u}_n \in \mathbb{R}^m$. The weight matrices are $\mathbf{W}_{1,2,z,y} \in \mathbb{R}^{d \times d}$ and $\mathbf{V}_{1,2,z,y} \in \mathbb{R}^{d \times m}$ and the bias vectors are $\mathbf{b}_{1,2,z,y} \in \mathbb{R}^d$. We also augment LEM (3) with a linear *output state* $\omega_n \in \mathbb{R}^o$ with $\omega_n = \mathcal{W}_y \mathbf{y}_n$, and $\mathcal{W}_y \in \mathbb{R}^{o \times d}$.

## 3 RELATED WORK

We start by comparing our proposed model, LEM (3), to the widely used LSTM of Hochreiter & Schmidhuber (1997). Observe that $\boldsymbol{\Delta t}_n, \overline{\boldsymbol{\Delta t}}_n$ in (3) are similar in form to the *input, forget* and *output* gates in an LSTM (see **SM**§D), and that LEM (3) has exactly the same number of parameters (weights and biases) as an LSTM, for the same number of hidden units. Moreover, as detailed in **SM**§D, we show that by choosing very specific values of the LSTM gates and the $\boldsymbol{\Delta t}_n, \overline{\boldsymbol{\Delta t}}_n$ terms in LEM (3), the two models are equivalent. However, this analysis also reveals key differences between LEM (3) and LSTMs, as they are equivalent only under very stringent assumptions. In general, as the different gates in both LSTM and LEM (3) are *learned* from data, one can expect them to behave differently. Moreover in contrast to LSTM, LEM stems from a discretized ODE system (2), which endows it with (gradient) stable dynamics.

The use of *multiscale* neural network architectures in machine learning has a long history. An early example was provided in Hinton & Plaut (1987), who proposed a neural network with each connection having a fast changing weight for temporary memory and a slow changing weight for long-term learning. More recently, one can view convolutional neural networks as multiscale architectures for processing multiple *spatial* scales in data (Bai et al., 2020).

The use of ODE-based learning architectures has also received considerable attention in recent years with examples such as *continuous-time* neural ODEs (Chen et al., 2018; Queiruga et al., 2020; 2021) and their recurrent extensions ODE-RNNs (Rubanova et al., 2019), as well as RNNs based on discretizations of ODEs (Chang et al., 2018; Erichson et al., 2021; Chen et al., 2020; Lim et al., 2021; Rusch & Mishra, 2021a;b). In addition to the specific details of our archiecture, we differ from other discretized ODE-based RNNs in the explicit use of multiple (learned) scales in LEM.

## 4 RIGOROUS ANALYSIS OF LEM

**Bounds on hidden states.** The structure of LEM (3) allows us to prove (in **SM**§E.1) that its hidden states satisfy the following *pointwise bound*.

**Proposition 4.1.** *Denote $t_n = n\Delta t$ and assume that $\Delta t \le 1$. Further assume that the initial hidden states are $\mathbf{z}_0 = \mathbf{y}_0 \equiv 0$. Then, the hidden states $\mathbf{z}_n, \mathbf{y}_n$ of LEM (3) are bounded pointwise as,*

$$
\max_{1 \le i \le d} \max\{|\mathbf{z}_n^i|, |\mathbf{y}_n^i|\} \le \min\left(1, \overline{\Delta}\sqrt{t_n}\right), \quad \forall 1 \le n, \text{ with } \overline{\Delta} = \frac{1 + \Delta t}{\sqrt{2 - \Delta t}}.
\tag{4}
$$

**On the exploding and vanishing gradient problem.** For any $1 \le n \le N$, let $\mathbf{X}_n \in \mathbb{R}^{2d}$, denoted the *combined hidden state*, given by $\mathbf{X}_n = \left[\mathbf{z}_n^1, \mathbf{y}_n^1, \dots, \mathbf{z}_n^i, \mathbf{y}_n^i, \dots, \mathbf{z}_n^d, \mathbf{y}_n^d\right]$. For simplicity of the exposition, we consider a *loss function*: $\mathcal{E}_n = \frac{1}{2}\|\mathbf{y}_n - \overline{\mathbf{y}}_n\|^2$, with $\overline{\mathbf{y}}_n$ being the underlying *ground truth*. The training of our proposed model (3) entails computing *gradients* of the above loss function with respect to its underlying weights and biases $\theta \in \boldsymbol{\Theta} = [\mathbf{W}_{\mathbf{1,2,y,z}}, \mathbf{V}_{\mathbf{1,2,y,z}}, \mathbf{b}_{\mathbf{1,2,y,z}}]$,

at every step of the gradient descent procedure. Following Pascanu et al. (2013), one uses chain rule to show,

$$\frac{\partial \mathcal{E}_n}{\partial \theta} = \sum_{1 \leq k \leq n} \frac{\partial \mathcal{E}_n^{(k)}}{\partial \theta}, \quad \frac{\partial \mathcal{E}_n^{(k)}}{\partial \theta} = \frac{\partial \mathcal{E}_n}{\partial \mathbf{X}_n} \frac{\partial \mathbf{X}_n}{\partial \mathbf{X}_k} \frac{\partial^+ \mathbf{X}_k}{\partial \theta} \tag{5}$$

In general, for recurrent models, the partial gradient $\frac{\partial \mathcal{E}_n^{(k)}}{\partial \theta}$, which measures the contribution to the hidden state gradient at step $n$ arising from step $k$ of the model, can behave as $\frac{\partial \mathcal{E}_n^{(k)}}{\partial \theta} \sim \gamma^{n-k}$, for some $\gamma > 0$ Pascanu et al. (2013). If $\gamma > 1$, then the partial gradient grows *exponentially* in sequence length, for long-term dependencies $k << n$, leading to the exploding gradient problem. On the other hand, if $\gamma < 1$, then partial gradients decays *exponentially* for $k << n$, leading to the vanishing gradient problem. Thus, mitigation of the exploding and vanishing gradient problem entails deriving bounds on the gradients. We start with the following upper bound (proved in **SM**§E.2),

**Proposition 4.2.** *Let $\mathbf{z}_n, \mathbf{y}_n$ be the hidden states generated by LEM (3). We assume that $\Delta t << 1$ is chosen to be sufficiently small. Then, the gradient of the loss function $\mathcal{E}_n$ with respect to any parameter $\theta \in \mathbf{\Theta}$ is bounded as*

$$\left| \frac{\partial \mathcal{E}_n}{\partial \theta} \right| \leq (1 + \hat{\mathbf{Y}}) t_n + (1 + \hat{\mathbf{Y}}) \Gamma t_n^2, \quad \hat{Y} = \|\overline{\mathbf{y}}_n\|_\infty, \tag{6}$$

$$\eta = \max\{\|\mathbf{W}_1\|_\infty, \|\mathbf{W}_2\|_\infty, \|\mathbf{W}_z\|_\infty, \|\mathbf{W}_y\|_\infty\}, \quad \Gamma = 2(1+\eta)(1+3\eta)$$

If we choose the hyperparameter $\Delta t = \mathcal{O}(n^{-1})$ (see SM (17) for the order-notation), then one readily observes from (6) that the gradient $\partial_\theta \mathcal{E}_n$ is *uniformly bounded* for any sequence length $n$ and the exploding gradient problem is clearly mitigated for LEM (3). Even if one chooses $\Delta t = \mathcal{O}(n^{-s})$, for some $0 \leq s \leq 1$, we show in **SM** Remark E.1 that the gradient can only grow polynomially (e.g. as $\mathcal{O}(n)$ for $s = 1/2$), still mitigating the exploding gradient problem.

Following Pascanu et al. (2013), one needs a more precise characterization of the partial gradient $\partial_\theta \mathcal{E}_n^{(k)}$, for long-term dependencies, i.e., $k << n$, to show mitigation of the vanishing gradient problem. In **SM**§E.3, we state and prove proposition E.2, which provides a precise formula for the asymptotics of the partial gradient. Here, we illustrate this formula in a special case as a corollary,

**Proposition 4.3.** *Let $\mathbf{y}_n, \mathbf{z}_n$ be the hidden states generated by LEM (3) and the ground truth satisfy $\overline{\mathbf{y}}_n \sim \mathcal{O}(1)$. Then, for any $k << n$ (long-term dependencies) we have,*

$$\frac{\partial \mathcal{E}_n^{(k)}}{\partial \theta} = \mathcal{O}\left(\Delta t^{\frac{3}{2}}\right). \tag{7}$$

*Here, constants in $\mathcal{O}(\Delta t^{\frac{3}{2}})$ depend on only on $\eta$ (6) and $\overline{\eta} = \|\mathbf{W}_2\|_1$ and are independent of $n, k$.*

This formula (7) shows that although the partial gradient can be small, i.e., $\mathcal{O}(\Delta t^{\frac{3}{2}})$, it is in fact *independent of $k$*, ensuring that long-term dependencies contribute to gradients at much later steps and mitigating the vanishing gradient problem.

**Universal approximation of general dynamical systems.** The above bounds on hidden state gradients show that the proposed model LEM (3) mitigates the exploding and vanishing gradients problem. However, this by itself, does not guarantee that it can learn complicated and realistic input-output maps between sequences. To investigate the *expressivity* of the proposed LEM, we will show in the following proposition that it can approximate *any* dynamical system, mapping an input sequence $\mathbf{u}_n$ to an output sequence $\mathbf{o}_n$, of the (very) general form,

$$\boldsymbol{\phi}_n = \mathbf{f}\left(\boldsymbol{\phi}_{n-1}, \mathbf{u}_n\right), \quad \mathbf{o}_n = \mathbf{o}(\boldsymbol{\phi}_n), \quad \forall 1 \leq n \leq N, \tag{8}$$

with $\boldsymbol{\phi}_n \in \mathbb{R}^{d_h}, \mathbf{o}_n \in \mathbb{R}^{d_o}$ denoting the *hidden* and *output* states, respectively. The input signal is $\mathbf{u}_n \in \mathbb{R}^{d_u}$ and maps $\mathbf{f} : \mathbb{R}^{d_h} \times \mathbb{R}^{d_u} \mapsto \mathbb{R}^{d_h}$ and $\mathbf{o} : \mathbb{R}^{d_h} \mapsto \mathbb{R}^{d_o}$ are Lipschitz continuous. For simplicity, we set the initial state $\boldsymbol{\phi}_0 = 0$.

**Proposition 4.4.** *For all $1 \leq n \leq N$, let $\boldsymbol{\phi}_n, \mathbf{o}_n$ be given by the dynamical system (8) with input signal $\mathbf{u}_n$. Under the assumption that there exists a $R > 0$ such that $\max\{\|\boldsymbol{\phi}_n\|, \|\mathbf{u}_n\|\} < R$, for all $1 \leq n \leq N$, then for any given $\epsilon > 0$ there exists a LEM of the form (3), with hidden states $\mathbf{y}_n, \mathbf{z}_n \in \mathbb{R}^{d_y}$ and output state $\omega_n = \mathcal{W}_y \mathbf{y}_n \in \mathbb{R}^{d_o}$, for some $d_y$ such that the following holds,*

$$\|\mathbf{o}_n - \omega_n\| \leq \epsilon, \quad \forall 1 \leq n \leq N. \tag{9}$$

From this proposition, proved in **SM**§E.4, we conclude that, in principle, the proposed LEM (3) can approximate a very large class of dynamical systems.

**Universal approximation of multiscale dynamical systems.** While expressing a general form of input-output maps between sequences, the dynamical system (8) does not explicitly model dynamics at multiple scales. Instead, here we consider the following two-scale *fast-slow* dynamical system of the general form,

$$\boldsymbol{\phi}_n = \mathbf{f}(\boldsymbol{\phi}_{n-1}, \boldsymbol{\psi}_{n-1}, \mathbf{u}_n), \quad \boldsymbol{\psi}_n = \tau \mathbf{g}(\boldsymbol{\phi}_n, \boldsymbol{\psi}_{n-1}, \mathbf{u}_n), \quad \mathbf{o}_n = \mathbf{o}(\boldsymbol{\psi}_n). \tag{10}$$

Here, $0 < \tau << 1$ and $1$ are the slow and fast time scales, respectively. The underlying maps $(\mathbf{f}, \mathbf{g}) : \mathbb{R}^{d_h \times d_h \times d_u} \mapsto \mathbb{R}^{d_h}$ are Lipschitz continuous. In the following proposition, proved in **SM**§E.5, we show that LEM (3) can approximate (10) to desired accuracy.

**Proposition 4.5.** *For any $0 < \tau << 1$, and for all $1 \leq n \leq N$, let $\boldsymbol{\phi}_n, \boldsymbol{\psi}_n, \mathbf{o}_n$ be given by the two-scale dynamical system* (10) *with input signal $\mathbf{u}_n$. Under the assumption that there exists a $R > 0$ such that $\max\{\|\boldsymbol{\phi}_n\|, \|\boldsymbol{\psi}_n\|, \|\mathbf{u}_n\|\} < R$, for all $1 \leq n \leq N$, then for any given $\epsilon > 0$, there exists a LEM of the form* (3), *with hidden states $\mathbf{y}_n, \mathbf{z}_n \in \mathbb{R}^{d_y}$ and output state $\omega_n \in \mathbb{R}^{d_o}$ with $\omega_n = \mathcal{W}\mathbf{y}_n$ such that the following holds,*

$$\|\mathbf{o}_n - \omega_n\| \leq \epsilon, \quad \forall 1 \leq n \leq N. \tag{11}$$

*Moreover, the weights, biases and size (number of neurons) of the underlying LEM* (3) *are* independent *of the time-scale $\tau$.*

This argument can be readily generalized to more than two time scales (see **SM** Proposition E.4). Hence, we show that, in principle, the proposed model LEM (3) can approximate multiscale dynamical systems, with model size being *independent* of the underlying timescales. These theoretical results for LEM (3) point to the ability of this architecture to learn complicated multiscale input-output maps between sequences, while mitigating the exploding and vanishing gradients problem. Although useful prerequisities, these theoretical properties are certainly not sufficient to demonstrate that LEM (3) is efficient in practice. To do this, we perform several benchmark evaluations, and we report the results below.

## 5 EMPIRICAL RESULTS

We present a variety of experiments ranging from long-term dependency tasks to real-world applications as well as tasks which require high expressivity of the model. Details of the training procedure for each experiment can be found in **SM**§A. As competing models to LEM, we choose two different types of architectures—LSTMs and GRUs—as they are known to excel at expressive tasks such as language modeling and speech recognition, while not performing well on long-term dependency tasks, possibly due to the exploding and vanishing gradients problem. On the other hand, we choose state-of-the-art RNNs which are tailor-made to learn tasks with long-term dependencies. Our objective is to evaluate the performance of LEM and compare it with competing models. All code to reproduce our results can be found at **https://github.com/tk-rusch/LEM**.

**Very long adding problem.** We start with the well-known adding problem (Hochreiter & Schmidhuber, 1997), proposed to test the ability of a model to learn (very) long-term dependencies. The input is a two-dimensional sequence of length $N$, with the first dimension consisting of random numbers drawn from $\mathcal{U}([0, 1])$ and with two non-zero entries (both set to 1) in the second dimension, chosen at random locations, but one each in both halves of the sequence. The output is the sum of two numbers of the first dimension at positions, corresponding to the two 1 entries in the second dimension. We consider three very challenging cases, namely input sequences with length $N = 2000, 5000$ and $10000$. The results of LEM together with competing models including state-of-the-art RNNs, which are explicitly designed to solve long-term dependencies, are presented in Fig. 1. We observe in this figure that while baseline LSTM is not able to beat the baseline mean-square error of $0.167$ (the variance of the baseline output 1) in any of the three cases, a proper weight initialization for LSTM, the so-called *chrono*-initialization of Tallec & Ollivier (2018) leads to much better performance in all cases. For $N = 2000$, all other architectures (except baseline LSTM) beat the baseline convincingly. However for $N = 5000$, only LEM, chrono-LSTM and coRNN are able

to beat the baseline. In the extreme case of $N = 10000$, only LEM and chrono-LSTM are able to beat the baseline. Nevertheless, LEM outperforms chrono-LSTM by converging faster (in terms of number of training steps) and attaining a lower test MSE than chrono-LSTM in all three cases.

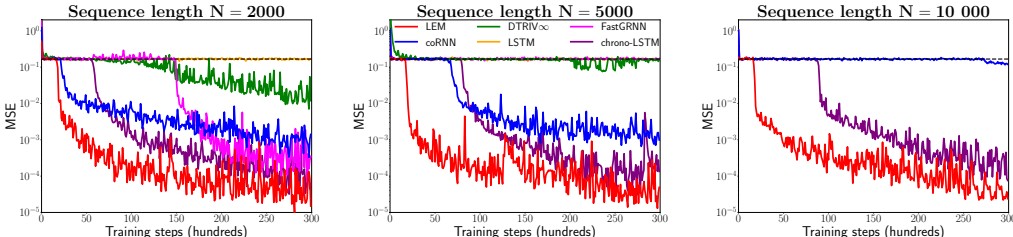

Figure 1: Results on the very long adding problem for LEM, coRNN, DTRIV$\infty$ (Casado, 2019), FastGRNN (Kusupati et al., 2018), LSTM and LSTM with chrono initialization (Tallec & Ollivier, 2018) based on three very long sequence lengths $N$, i.e., $N = 2000$, $N = 5000$ and $N = 10000$.

**Sequential image recognition.** We consider three experiments based on two widely-used image recognition data sets, i.e., MNIST (LeCun et al., 1998) and CIFAR-10 (Krizhevsky et al., 2009), where the goal is to predict the correct label after reading in the whole sequence. The first two tasks are based on MNIST images, which are flattened along the rows to obtain sequences of length $N = 784$. In sequential MNIST (sMNIST), the sequences are fed to the model one pixel at a time in streamline order, while in permuted sequential MNIST (psMNIST), a fixed random permutation is applied to the sequences, resulting in much longer dependency than for sMNIST. We also consider the more challenging noisy CIFAR-10 (nCIFAR-10) experiment (Chang et al., 2018), where CIFAR-10 images are fed to the model row-wise and flattened along RGB channels, resulting in 96-dimensional sequences, each of length 32. Moreover, a random noise padding is applied after the first 32 inputs to produce sequences of length $N = 1000$. Hence, in addition to classifying the underlying image, a model has to store this result for a long time. In Table 1, we present the results for LEM on the three tasks together with other SOTA RNNs, which were explicitly designed to solve long-term dependency tasks, as well as LSTM and GRU baselines. We observe that LEM outperforms all other methods on sMNIST and nCIFAR-10. Additionally on psMNIST, LEM performs as well as coRNN, which has been SOTA among single-layer RNNs on this task.

Table 1: Test accuracies on sMNIST, psMNIST and nCIFAR-10, where $M$ denotes the total number of parameters of the corresponding model. Results of other models are taken from the respective original paper referenced in the main text, except that the results for LSTM are taken from Helfrich et al. (2018), for GRU from Chang et al. (2017) and the results indicated by * are added by us.

| Model | MNIST | | | CIFAR-10 | |
|---|---|---|---|---|---|
| | sMNIST | psMNIST | # units / $M$ | nCIFAR-10 | # units / $M$ |
| GRU | 99.1% | 94.1% | 256 / 201k | 43.8%* | 128 / 88k |
| LSTM | 98.9% | 92.9% | 256 / 267k | 11.6% | 128 / 116k |
| chrono-LSTM | 98.9%* | 94.6%* | 128 / 68k | 55.9%* | 128 / 116k |
| anti.sym. RNN | 98.0% | 95.8% | 128 / 10k | 48.3% | 256 / 36k |
| Lipschitz RNN | 99.4% | 96.3% | 128 / 34k | 57.4% | 128 / 46k |
| expRNN | 98.4% | 96.2% | 360 / 69k | 52.9%* | 360 / 103k |
| coRNN | 99.3% | **96.6**% | 128/ 34k | 59.0% | 128 / 46k |
| **LEM** | **99.5**% | **96.6**% | 128 / 68k | **60.5**% | 128 / 116k |

**EigenWorms: Very long sequences for genomics classification.** The goal of this task (Bagnall et al., 2018) is to classify worms as belonging to either the wild-type or four different mutants, based on 259 very long sequences (length $N = 17984$) measuring the motion of a worm. In addition to the nominal length, it was empirically shown in Rusch & Mishra (2021b) that the EigenWorms sequences exhibit actual very long-term dependencies (i.e., longer than 10k).

Table 2: Test accuracies on EigenWorms using 5 re-trainings of each best performing network (based on the validation set), where all other results are taken from Rusch & Mishra (2021b) except that the NRDE result is taken from Morrill et al. (2021) and the results indicated by $*$ are added by us.

| Model | test accuracy | # units | # params |
|---|---|---|---|
| NRDE | $83.8\% \pm 3.0\%$ | 32 | 35k |
| expRNN | $40.0\% \pm 10.1\%$ | 64 | 2.8k |
| IndRNN (2 layers) | $49.7\% \pm 4.8\%$ | 32 | 1.6k |
| LSTM | $38.5\% \pm 10.1\%*$ | 32 | 5.3k |
| BiLSTM+1d-conv | $40.5\% \pm 7.3\%*$ | 22 | 5.8k |
| chrono-LSTM | $82.6\% \pm 6.4\%*$ | 32 | 5.3k |
| coRNN | $86.7\% \pm 3.0\%$ | 32 | 2.4k |
| UnICORNN (2 layers) | $90.3\% \pm 3.0\%$ | 32 | 1.5k |
| **LEM** | $\mathbf{92.3\% \pm 1.8\%}$ | 32 | 5.3k |

Following Morrill et al. (2021) and Rusch & Mishra (2021b), we divide the data into a train, validation and test set according to a $70\%, 15\%, 15\%$ ratio. In Table 2, we present results for LEM together with other models. As the validation and test sets, each consist of only 39 sequences, we report the mean (and standard deviation of) accuracy over 5 random initializations to rule out lucky outliers. We observe from this table that LEM outperforms all other methods, even the 2-layer UnICORNN architecture, which has been SOTA on this task.

**Healthcare application: Heart-rate prediction.** In this experiment, one predicts the heart rate from a time-series of measured PPG data, which is part of the TSR archive (Tan et al., 2020) and has been collected at the Beth Isreal Deaconess medical center. The data set, consisting of 7949 sequences, each of length $N = 4000$, is divided into a train, validation and test set according to a 70%,15%,15% ratio, (Morrill et al., 2021; Rusch & Mishra, 2021b). The results, presented in Table 3, show that LEM outperforms the other competing models, including having a test $L^2$ error of 35% less than the SOTA UnICORNN.

Table 3: Test $L^2$ error on heart-rate prediction using PPG data. All results are obtained by running the same code and using the same fine-tuning protocol.

| Model | test $L^2$ error | # units | # params |
|---|---|---|---|
| LSTM | 9.93 | 128 | 67k |
| chrono-LSTM | 3.31 | 128 | 67k |
| expRNN | 1.63 | 256 | 34k |
| IndRNN (3 layers) | 1.94 | 128 | 34k |
| coRNN | 1.61 | 128 | 34k |
| UnICORNN (3 layers) | 1.31 | 128 | 34k |
| **LEM** | **0.85** | 128 | 67k |

**Multiscale dynamical system prediction.** The FitzHugh-Nagumo system (Fitzhugh, 1955)

$$v' = v - \frac{v^3}{3} - w + I_{\text{ext}}, \quad w' = \tau(v + a - bw), \quad (12)$$

is a prototypical model for a two-scale fast-slow nonlinear dynamical system, with fast variable $v$ and slow variable $w$ and $\tau << 1$ determining the slow-time scale. This *relaxation-oscillator* is an approximation to the Hodgkin-Huxley model (Hodgkin & Huxley, 1952) of neuronal action-potentials under an external signal $I_{\text{ext}} \geq 0$. With $\tau = 0.02$, $I_{\text{ext}} = 0.5$, $a = 0.7$, $b = 0.8$ and initial data $(v_0, w_0) = (c, 0)$, with $c$ randomly drawn from $\mathcal{U}([-1, 1])$, we numerically approximate (12) with the explicit Runge-Kutta method of order $5(4)$ in the interval $[0, 400]$ and generate 128 training and validation and 1024 test sequences, each of length $N = 1000$, to complete the data set. The results, presented in Table 4, show that LEM not only outperforms LSTM by a factor of 6 but also all other methods including coRNN, which is tailormade for oscillatory time-series. This reinforces our theory by demonstrating efficient approximation of multiscale dynamical systems with LEM.

Table 4: Test $L^2$ error on FitzHugh-Nagumo system prediction. All results are obtained by running the same code and using the same fine-tuning protocol.

| Model | error ($\times 10^{-2}$) | # units | # params |
|---|---|---|---|
| LSTM | 1.2 | 16 | 1k |
| expRNN | 2.3 | 50 | 1k |
| LipschitzRNN | 1.8 | 24 | 1k |
| FastGRNN | 2.2 | 34 | 1k |
| coRNN | 0.4 | 24 | 1k |
| **LEM** | **0.2** | 16 | 1k |

**Google12 (V2) keyword spotting.** The Google Speech Commands data set V2 (Warden, 2018) is a widely used benchmark for keyword spotting, consisting of 35 words, sampled at a rate of 16 kHz from 1 second utterances of 2618 speakers. We focus on the 12-label task (Google12) and follow the pre-defined splitting of the data set into train/validation/test sets and test different sequential models. In order to ensure comparability of different architectures, we do not use performance-enhancing tools such as convolutional filtering or multi-head attention. From Table 5, we observe that both LSTM and GRU, widely used models in this context, perform very well with a test accuracy of around 95%. Nevertheless, LEM is able to outperform both on this task and provides the best performance.

Table 5: Test accuracies on Google12. All results are obtained by running the same code and using the same fine-tuning protocol.

| Model | test accuracy | # units | # params |
|---|---|---|---|
| tanh-RNN | 73.4% | 128 | 27k |
| anti.sym. RNN | 90.2% | 128 | 20k |
| LSTM | 94.9% | 128 | 107k |
| GRU | 95.2% | 128 | 80k |
| FastGRNN | 94.8% | 128 | 27k |
| expRNN | 92.3% | 128 | 19k |
| coRNN | 94.7% | 128 | 44k |
| **LEM** | **95.7**% | 128 | 107k |

**Language modeling: Penn Tree Bank corpus.** Language modeling with the widely used small scale Penn Treebank (PTB) corpus (Marcus et al., 1993), preprocessed by Mikolov et al. (2010), has been identified as an excellent task for testing the expressivity of recurrent models (Kerg et al., 2019). To this end, in Table 6, we report the results of different architectures, with a similar number of hidden units, on the PTB char-level task and observe that RNNs, designed explicitly for learning long-term dependencies, perform significantly worse than LSTM and GRU. On the other hand, LEM is able to outperform both LSTM and GRU on this task by some margin (a test bpc of 1.25 in contrast with approximately a bpc of 1.36). In fact, LEM provides the smallest test bpc among all reported single-layer recurrent models on this task, to the best of our knowledge. This superior performance is further illustrated in Table 7, where the test perplexity for different models on the PTB word-level task is presented. We observe that not only does LEM significantly outperform (by around 40%) LSTM, but it also provides again the best performance among all single layer recurrent models, including the recently proposed TARNN (Kag & Saligrama, 2021). Moreover, the single-layer results for LEM are better than reported results for multi-layer LSTM models, such as in Gal & Ghahramani (2016) (2-layer LSTM, 1500 units each: 75.2 test perplexity) or Bai et al. (2018) (3-layer LSTM, 700 units each: 78.93 test perplexity).

## 6 DISCUSSION

The design of a gradient-based model for processing sequential data that can learn tasks with long-term dependencies while retaining the ability to learn complicated sequential input-output maps is

Table 6: Test bits-per-character (bpc) on PTB character-level for single layer LEM and other single layer RNN architectures. Other results are taken from the papers cited accordingly in the table, while the results for coRNN are added by us.

| Model | test bpc | # units | # params |
|---|---|---|---|
| anti.sym RNN (Erichson et al., 2021) | 1.60 | 1437 | 1.3M |
| Lipschitz RNN (Erichson et al., 2021) | 1.42 | 764 | 1.3M |
| expRNN (Kerg et al., 2019) | 1.51 | 1437 | 1.3M |
| coRNN | 1.46 | 1024 | 2.3M |
| nnRNN (Kerg et al., 2019) | 1.47 | 1437 | 1.3M |
| LSTM (Krueger et al., 2017) | 1.36 | 1000 | 5M |
| GRU (Bai et al., 2018) | 1.37 | 1024 | 3M |
| **LEM** | **1.25** | 1024 | 5M |

Table 7: Test perplexity on PTB word-level for single layer LEM and other single layer RNN architectures.

| Model | test perplexity | # units | # params |
|---|---|---|---|
| Lipschitz RNN (Erichson et al., 2021) | 115.4 | 160 | 76k |
| FastRNN (Kag & Saligrama, 2021) | 115.9 | 256 | 131k |
| LSTM (Kag & Saligrama, 2021) | 116.9 | 256 | 524k |
| SkipLSTM (Kag & Saligrama, 2021) | 114.2 | 256 | 524k |
| TARNN (Kag & Saligrama, 2021) | 94.6 | 256 | 524k |
| **LEM** | **72.8** | 256 | 524k |

very challenging. In this paper, we have proposed *Long Expressive Memory* (LEM), a novel recurrent architecture, with a suitable time-discretization of a specific multiscale system of ODEs (2) serving as the circuit to the model. By a combination of theoretical arguments and extensive empirical evaluations on a diverse set of learning tasks, we demonstrate that LEM is able to learn long-term dependencies while retaining sufficient expressivity for efficiently solving realistic learning tasks.

It is natural to ask why LEM performs so well. A part of the answer lies in the mitigation of the exploding and vanishing gradients problem. Proofs for gradient bounds (6),(7) reveal a key role played by the hyperparameter $\Delta t$. We observe from **SM** Table 8 that small values of $\Delta t$ might be needed for problems with very long-term dependencies, such as the EigenWorms dataset. On the other hand, no tuning of the hyperparameter $\Delta t$ is necessary for several tasks such as language modeling, keyword spotting and dynamical systems prediction and a default value of $\Delta t = 1$ yielded very good performance. The role and choice of the hyperparameter $\Delta t$ is investigated extensively in **SM**§B.1. However, mitigation of exploding and vanishing gradients problem alone does not explain high expressivity of LEM. In this context, we proved that LEMs can approximate a very large class of multiscale dynamical systems. Moreover, we provide experimental evidence in **SM**§B.2 to observe that LEM not only expresses a range of scales, as it is designed to do, but also these scales contribute proportionately to the resulting multiscale dynamics. Furthermore, empirical results presented in **SM**§B.2 show that this ability to represent multiple scales correlates with the high accuracy of LEM. We believe that this combination of gradient stable dynamics, specific model structure, and its multiscale resolution can explain the observed performance of LEM.

We conclude with a comparison of LEM and the widely-used gradient-based LSTM model. In addition to having exactly the same number of parameters for the same number of hidden units, our experiments show that LEMs are better than LSTMs on expressive tasks such as keyword spotting and language modeling, while also providing significantly better performance on long-term dependencies. This robustness of the performance of LEM with respect to sequence length paves the way for its application to learning many different sequential data sets where competing models might not perform satisfactorily.

ACKNOWLEDGEMENTS.

The research of TKR and SM was performed under a project that has received funding from the European Research Council (ERC) under the European Union's Horizon 2020 research and innovation programme (grant agreement No. 770880). NBE and MWM would like to acknowledge IARPA (contract W911NF20C0035), NSF, and ONR for providing partial support of this work. Our conclusions do not necessarily reflect the position or the policy of our sponsors, and no official endorsement should be inferred.

The authors thank Dr. Ivo Danihelka (DeepMind) for pointing out that the hidden states for LEM satisfy the maximum principles (19), (20).

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

**Supplementary Material for:**
Long Expressive Memory for Sequence Modeling



## A  TRAINING DETAILS

All experiments were run on CPU, namely Intel Xeon Gold 5118 and AMD EPYC 7H12, except for Google12, PTB character-level and PTB word-level, which were run on a GeForce RTX 2080 Ti GPU. All weights and biases of LEM (3) are initialized according to $\mathcal{U}(-1/\sqrt{d}, 1/\sqrt{d})$, where $d$ is the number of hidden units.

Table 8: Rounded hyperparameters of the best performing LEM architecture for each experiment. If no value is given for $\Delta t$, it means that $\Delta t$ is fixed to 1 and no fine-tuning is performed on this hyperparameter.

| experiment | learning rate | batch size | $\Delta t$ |
|---|---|---|---|
| Adding ($N = 10000$) | $2.6 \times 10^{-3}$ | 50 | $2.42 \times 10^{-2}$ |
| sMNIST | $1.8 \times 10^{-3}$ | 128 | $2.1 \times 10^{-1}$ |
| psMNIST | $3.5 \times 10^{-3}$ | 128 | $1.9 \times 10^{0}$ |
| nCIFAR-10 | $1.8 \times 10^{-3}$ | 120 | $9.5 \times 10^{-1}$ |
| EigenWorms | $2.3 \times 10^{-3}$ | 8 | $1.6 \times 10^{-3}$ |
| Healthcare | $1.56 \times 10^{-3}$ | 32 | $1.9 \times 10^{-1}$ |
| FitzHugh-Nagumo | $9.04 \times 10^{-3}$ | 32 | / |
| Google12 | $8.9 \times 10^{-4}$ | 100 | / |
| PTB character-level | $6.6 \times 10^{-4}$ | 128 | / |
| PTB word-level | $6.8 \times 10^{-4}$ | 64 | / |

The hyperparameters are selected based on a random search algorithm, where we present the rounded hyperparameters for the best performing LEM model (*based on a validation set*) on each task in Table 8.

We base the training for the PTB experiments on the following language modelling code: https://github.com/deepmind/lamb, where we fine-tune, based on a random search algorithm, only the learning rate, input-, output- and state-dropout, $L^2$-penalty term and the maximum gradient norm.

We train LEM for 100 epochs on sMNIST, psMNIST and nCIFAR-10, after which we decrease the learning rate by a factor of 10 and proceed training for 20 epochs. Moreover, we train LEM for 50, 60 as well as 400 epochs on EigenWorms, Google12 and FitzHugh-Nagumo. We decrease the learning rate by a factor of 10 after 50 epochs on Google12. On the Healthcare task, we train LEM for 250 epochs, after which we decrease the learning rate by a factor of 10 and proceed training for 250 epochs.

## B  FURTHER EXPERIMENTAL RESULTS

### B.1  ON THE CHOICE OF THE HYPERPARAMETER $\Delta t$.

The hyperparameter $\Delta t$ in LEM (3) measures the maximum allowed (time) step in the discretization of the multi-scale ODE system (2). In propositions 4.1, 4.2 and 4.3, this hyperparameter $\Delta t$ plays a key role in the bounds on the hidden states (4) and their gradients (6). In particular, setting $\Delta t = \mathcal{O}(N^{-1})$ will lead to hidden states and gradients, that are bounded uniformly with respect to the underlying sequence length $N$. However, these upper bounds on the hidden states and gradients account for *worst-case* scenarios and can be very pessimistic for the problem at hand.

Thus, in practice, we determine $\Delta t$ through a hyperparameter tuning procedure as described in section A. To this end, we perform a random search within $\Delta t < 2$ and present the resulting optimal values of $\Delta t$ for each of the considered data sets in Table 8. From this table, we observe that for data sets such as PTB, FitzHugh-Nagumo and Google 12 we do not need any tuning of $\Delta t$ and a

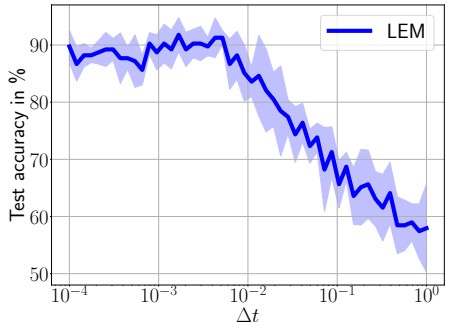 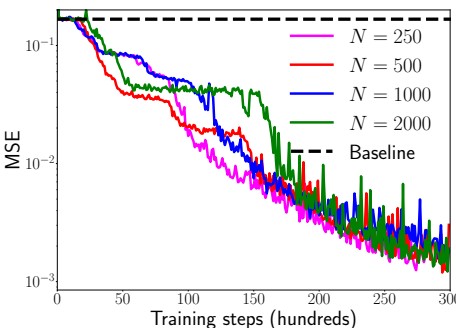

Figure 2: Sensitivity study on hyperparameter $\Delta t$ in (3) using the EigenWorms experiment.

Figure 3: Average (over ten different initializations each) test mean-square error on the adding problem of LEM for different sequence lengths $N$, where the hyperparameter $\Delta t$ of LEM (3) is fixed to $\Delta t = 1/\sqrt{N}$.

default value of $\Delta t = 1$ resulted in very good empiricial performance. On the other data sets such as sMNIST, nCIFAR-10 and the healthcare example, where the sequence length ($N = \mathcal{O}(10^3)$) is larger, we observe that values of $\Delta t \approx 0.1$ yielded the best performance. The notable exception to this was for the EigenWorms data set, with a very long sequence length of $N = 17984$ as well as demonstrated very long range dependencies in the data, see Rusch & Mishra (2021b). Here, a value of $\Delta t = 1.6 \times 10^{-3}$ resulted in the best observed performance. To further investigate the role of the hyperparameter $\Delta t$ in the EigenWorms experiment, we perform a sensitivity study where the value of $\Delta t$ is varied and the corresponding accuracy of the trained LEM is observed. The results of this sensitivity study are presented in Fig. 2, where we plot the test accuracy (Y-axis) vs. the value of $\Delta t$ (X-axis). From this figure, we observe that the accuracy is rather poor for $\Delta t \approx 1$ but improves monotonically as $\Delta t$ is reduced till a value of approximately $10^{-2}$, after which it saturates. Thus, in this case, a value of $\Delta t = \mathcal{O}(N^{-\frac{1}{2}})$ (for sequence length $N$) suffices to yield the best empirical performance.

Given this observation, we further test whether $\Delta t = \mathcal{O}(N^{-\frac{1}{2}})$ suffices for other problems with long-term dependencies. To this end, we consider the adding problem and vary the input sequence length by an order of magnitude, i.e., from $N = 250$ to $N = 2000$. The value of $\Delta t$ is now fixed at $\Delta t = \frac{1}{\sqrt{N}}$ and the resulting test loss (Y-axis) vs the number of training steps (X-axis) is plotted in Fig. 3. We see from this figure that this value of $\Delta t$ sufficed to yield very small average test errors for this problem for all considered sequence lengths $N$. Thus, empirically a value of $\Delta t$ in the range $\frac{1}{\sqrt{N}} \leq \Delta t \leq 1$ yields very good performance.

Even if we set $\Delta t = \mathcal{O}(\frac{1}{\sqrt{N}})$, it can happen for very long sequences $N >> 1$ that the gradient can be quite small from the gradient asymptotic formula (7). This might lead to saturation in training, resulting in long training times. However, we do not observe such long training times for very long sequence lengths in our experiment. To demonstrate this, we again consider Fig. 3 where the number of training steps (X-axis) is plotted for sequence lengths that vary an order of magnitude. The figure clearly shows that the approximately the same number of training steps are needed to attain a low test error, irrespective of the sequence length. This is further buttressed in Fig. 1, where similar number of training steps where needed for obtaining the same very low test error, even for long sequence lengths, with $N$ up to 10000. Moreover, from section A, we see that the number of epochs for different data sets is independent of the sequence length. For instance, only 50 epochs were necessary for EigenWorms with a sequence length of $N = 17984$ and $\Delta t = 1.6 \times 10^{-3}$ whereas 400 epochs were required for the FitzHugh-Nagumo system with a $\Delta t = 1$.

### B.2 Multiscale Behavor of LEM.

LEM (3) is designed to represent multiple scales, with terms $\boldsymbol{\Delta t}_n, \overline{\boldsymbol{\Delta t}}_n$ being explicitly designed to learn possible multiple scales. In the following, we will investigate if in practice, LEM learns multiple scales and uses them to yield the observed superior empirical performance with respect to competing models.

To this end, we start by recalling the proposition 4.5 where we showed that in principle, LEM can learn the two underlying timescales of a *fast-slow* dynamical system (see proposition E.4 for a similar result for the universal approximation of a $r$-time scale (with $r \geq 2$) dynamical system with LEM). Does this hold in practice ? To further investigate this issue, we consider the FitzHugh-Nagumo dynamical system (12) which serves as a prototype for a two-scale dynamical system. We consider this system (12) with the two time-scales being $\tau = 0.02$ and 1 and train LEM for this system. In Fig. 4, we plot the empirical histogram that bins the ranges of learned scales $\boldsymbol{\Delta t}_n, \overline{\boldsymbol{\Delta t}}_n \leq \Delta t = 2$ (for all $n$ and $d$) and counts the number of occurrences of $\boldsymbol{\Delta t}_n, \overline{\boldsymbol{\Delta t}}_n$ in each bin. From this figure, we observe that there is a clear concentration of learned scales around the values 1 and $\tau = 0.02$, which exactly correspond to the underlying fast and slow time scales. Thus, for this model problem, LEM is exactly learning what it is designed to do and is able to learn the underlying time scales for this particular problem.

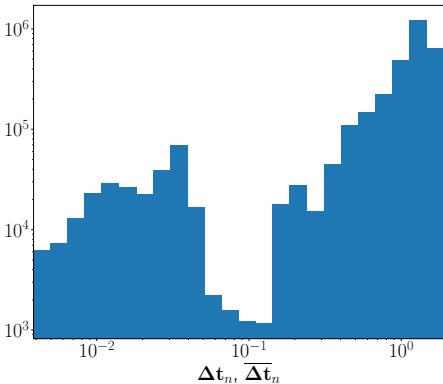

Figure 4: Histogram of $(\boldsymbol{\Delta t}_n)_i$ and $(\overline{\boldsymbol{\Delta t}}_n)_i$ for all $n = 1, \ldots, N$ and $i = 1, \ldots, d$ of LEM (3) after training on the FitzHugh-Nagumo fast-slow system (12) using $\Delta t = 2$.

Nevertheless, one might argue that these learnable mutliple scales $\boldsymbol{\Delta t}_n, \overline{\boldsymbol{\Delta t}}_n$ are not necessary and a single scale would suffice to provide good empirical performance. We check this possibility on the FitzHugh-Nagumo data set by simply setting $\boldsymbol{\Delta t}_n, \overline{\boldsymbol{\Delta t}}_n \equiv \Delta t \boldsymbol{1}$ (with $\boldsymbol{1}$ being the vector with all entries set to 1), for all $n$ and tuning the hyperparameter $\Delta t$. The comparative results are presented in Table 9. We see from this table by not allowing for learnable $\boldsymbol{\Delta t}_n, \overline{\boldsymbol{\Delta t}}_n$ and simply setting them to a single scale parameter $\Delta t$ and tuning this parameter only leads to results that are comparable to the baseline LSTM model. On the other hand, learning $\boldsymbol{\Delta t}_n, \overline{\boldsymbol{\Delta t}}_n$ resulted in an error that is a factor of 6 less than the baseline LSTM test error. Thus, we demonstrate the importance of the ability of the proposed LEM model to learn multiple scales in this example.

Table 9: Test $L^2$ error on FitzHugh-Nagumo system prediction.

| Model | error ($\times 10^{-2}$) | # units | # params |
|---|---|---|---|
| LSTM | 1.2 | 16 | 1k |
| LEM w/o multi-scale | 1.1 | 16 | 1k |
| LEM | 0.2 | 16 | 1k |

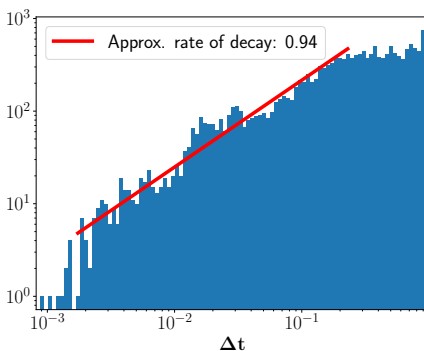 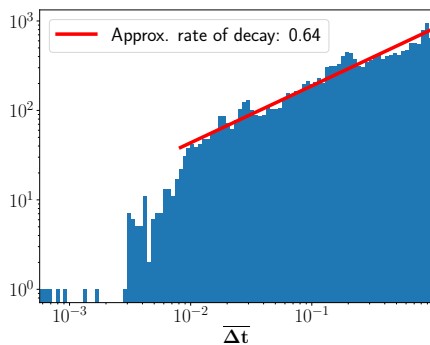

Figure 5: Histogram of $(\Delta\mathbf{t}_n)_i$ and $(\overline{\Delta\mathbf{t}}_n)_i$ for all $n = 1, \ldots, N$ and $i = 1, \ldots, d$ of LEM (3) after training on the Google12 data set

Hence, the multiscale resolution of LEM seems essential for the fast-slow dynamical system. Does this multiscale resolution also appear for other datasets and can it explain aspects of the observed empirical performance ? To this end, we consider the Google12 Keyword spotting data set and start by pointing out that given the spatial (with respect to input dimension $d$) and temporal (with respect to sequence length $N$) heterogeneities, a priori, it is unclear if the underlying data has a multiscale structure. We plot the empirical histograms of $\Delta\mathbf{t}_n, \overline{\Delta\mathbf{t}}_n$ in Fig. 5 to observe that even for this problem, the terms $\Delta\mathbf{t}_n, \overline{\Delta\mathbf{t}}_n$ are expressed over a range of scales, amounting to $2 - 3$ orders of magnitude. Thus, a range of scales are present in the trained LEM even for this example, but do they affect the empirical performance of LEM ? We investigate this question by performing an ablation study and reporting the results in Fig. 6. In this study, we clip the values of $\Delta\mathbf{t}_n, \overline{\Delta\mathbf{t}}_n$ to lie within the range $[2^{-i}, 1]$, for $i = 0, 1, \ldots, 7$ and plot the statistics of the observed test accuracy of LEM. We observe from Fig. 6 that by clipping $\Delta\mathbf{t}_n, \overline{\Delta\mathbf{t}}_n$ to lie near the default (single scale) value of 1 results in very poor empirical performance of an accuracy of $\approx 65\%$. Then the accuracy jumps to around $90\%$ when an order of magnitude range for $\Delta\mathbf{t}_n, \overline{\Delta\mathbf{t}}_n$ is considered, before monotonically and slowly increasing to yield the best empirical performance for the largest range of values of $\Delta\mathbf{t}_n, \overline{\Delta\mathbf{t}}_n$, considered in this study. A closer look at the empirical histograms plotted in Fig. 5 reveal that the proportion of occurrences of $\Delta\mathbf{t}_n, \overline{\Delta\mathbf{t}}_n$ decays as a *power law*, and not exponentially, with respect to the scale amplitude. This, together with results presented in Fig. 6 suggest that not only do a range of scales occur in learned $\Delta\mathbf{t}_n, \overline{\Delta\mathbf{t}}_n$, the small scales also contribute proportionately to the dynamics and enable the increase in performance shown in Fig. 6.

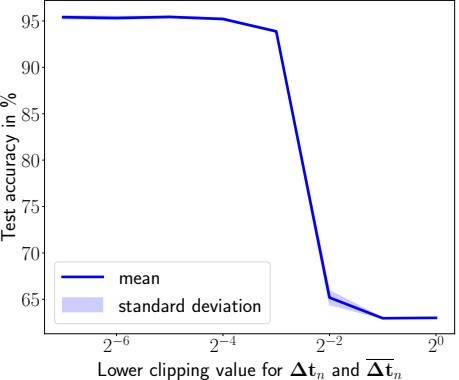

Figure 6: Average (and standard deviation of) test accuracies of 5 runs each for LEM on Google12, where $\Delta\mathbf{t}_n$ and $\overline{\Delta\mathbf{t}}_n$ in (3) are clipped to the ranges $[\frac{1}{2^i}, 1]$ for $i = 0, \ldots, 7$ during training.

Finally, in Fig. 7, we plot the empirical histograms of $\mathbf{\Delta t}_n$ and $\overline{\mathbf{\Delta t}}_n$ for the learned LEM on the sMNIST data set to observe that again a range of scales are observed and the observed occurrences of $\mathbf{\Delta t}_n$ and $\overline{\mathbf{\Delta t}}_n$ at each scale decays as a power law with respect to scale amplitude. Hence, we have sufficient empirical evidence to claim that the multi-scale resolution of LEM seems essential to its observed performance. However, further investigation is required to elucidate the precise mechanisms through this multiscale resolution enables superior performance, particularly on problems where the multiscale structure of the underlying data may not be explicit.

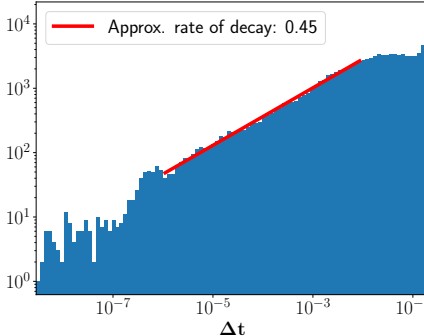 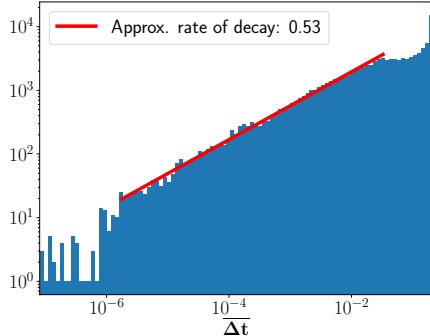

Figure 7: Histogram of $(\mathbf{\Delta t}_n)_i$ and $(\overline{\mathbf{\Delta t}}_n)_i$ for all $n = 1, \ldots, N$ and $i = 1, \ldots, d$ of LEM (3) after training on the sMNIST data set

### B.3 ON GRADIENT-STABLE INITIALIZATION.

Specialized weight initialization is a popular tool to increase the performance of RNNs on long-term dependencies tasks. One particular approach is the so-called chrono initialization (Tallec & Ollivier, 2018) for LSTMs, where all biases are set to zero except for the bias of the forget gate as well as the input gate ($\mathbf{b}_f$ and $\mathbf{b}_i$ in the LSTM (15)), which are sampled from

$$\mathbf{b}_f \sim \log(\mathcal{U}[1, T_{\max} - 1])$$
$$\mathbf{b}_i = -\mathbf{b}_f,$$

where $T_{\max}$ denotes the maximal temporal dependency of the underlying sequential data. We can see in Table 2 that the chrono initialization significantly improves the performance of LSTM on the EigenWorms task. Hence, we are interested in extending the chrono initialization to LEMs. One possible manner for doing this is as follows: Initialize all biases of LEM to zero except for $\mathbf{b}_1$ in (3), which is sampled from

$$\mathbf{b}_1 \sim -\log(\mathcal{U}[1, T_{\max}\Delta t - 1]).$$

Table 10: Test accuracies on EigenWorms using 5 re-trainings of each best performing network (based on the validation set), where we train LSTM and LEM with and without chrono intialization, as well as LEM without chrono initialization but with tuned $\Delta t$.

| Model | test accuracy | # units | # params | chrono | tuning $\Delta t$ |
|-------|--------------|---------|----------|--------|-------------------|
| LSTM | $38.5\% \pm 10.1\%$ | 32 | 5.3k | NO | / |
| LSTM | $82.6\% \pm 6.4\%$ | 32 | 5.3k | YES | / |
| LEM | $57.9\% \pm 7.7\%$ | 32 | 5.3k | NO | NO |
| LEM | $88.2\% \pm 6.9\%$ | 32 | 5.3k | YES | NO |
| LEM | $92.3\% \pm 1.8\%$ | 32 | 5.3k | NO | YES |

We test the chrono initialization for LEM on the EigenWorms dataset, where we train LEM (without tuning $\Delta t$, i.e., setting $\Delta t = 1$), with and without chrono initialization. We provide the results in Table 10, where we show again the results of LSTM with and without chrono initialization as well as the LEM result with tuned $\Delta t$ and without chrono initialization from Table 2 for comparison. We

see from Table 10 that when $\Delta t$ is fixed to 1, the chrono initialization significantly improves the result of LEM. However, if we tune $\Delta t$, but do not use the chrono initialization, we significantly improve the performance of LEM again. We further remark that tuning $\Delta t$ as well as using chrono initialization for LEM does not improve the results obtained with simply tuning $\Delta t$ in LEM. Thus, we conclude that chrono initialization can successfully be adapted to LEM. However, tuning $\Delta t$ (which controls the gradients) is still advisable in order to obtain the best possible results.

## C  RELATION BETWEEN LEM AND THE HODGKIN-HUXLEY EQUATIONS

We observe that the multiscale ODEs (2), on which LEM is based, are a special case of the following ODE system,

$$\frac{d\mathbf{z}}{dt} = \mathbf{F}_z\left(\mathbf{y},t\right) - \mathbf{G}_z(\mathbf{y},t) \odot \mathbf{z}, \quad \frac{d\mathbf{y}}{dt} = \mathbf{F}_y\left(\mathbf{z},t\right) \odot \mathbf{H}\left(\mathbf{y},t\right) - \mathbf{G}_y(\mathbf{y},t) \odot \mathbf{y}. \tag{13}$$

As remarked in the main text, it turns out the well-known Hodgkin-Huxley equations Hodgkin & Huxley (1952), modeling the the dynamics of the action potential of a biological neuron can also be written down in the abstract form (13), with $d_y = 1$, $d_z = 3$ and the variables $\mathbf{y} = y$ modeling the voltage and $\mathbf{z} = (z_1, z_2, z_3)$ modeling the concentration of Potassium activation, Sodium activation and Sodium inactivation channels.

The exact form of the different functions in (13) for the Hodgkin-Huxley equations is given by,

$$\begin{aligned}
\mathbf{F}_z(y) &= \left(\alpha_1(y), \alpha_2(y), \alpha_3(y)\right), \\
\mathbf{G}_z(y) &= \left(\alpha_1(y) + \beta_1(y), \alpha_2(y) + \beta_2(y), \alpha_3(y) + \beta_3(y)\right), \\
\alpha_1(y) &= \frac{0.01(10 + \hat{y} - y)}{e^{\frac{10+\hat{y}-y}{10}} - 1}, \quad \alpha_2(y) = \frac{0.1(25 + \hat{y} - y)}{e^{\frac{25+\hat{y}-y}{10}} - 1}, \quad \alpha_3(y) = 0.07 e^{\frac{\hat{y}-y}{20}}, \\
\beta_1(y) &= 0.125 e^{\frac{\hat{y}-y}{80}}, \quad \beta_2(y) = 4 e^{\frac{\hat{y}-y}{18}}, \quad \beta_3(y) = \frac{1}{1 + e^{1 + \frac{\hat{y}-y}{10}}}, \\
\mathbf{F}_y(z,t) &= u(t) + z_1^4 + z_2^3 z_3, \quad \mathbf{H}(y) = c_1(\bar{y} - y) + c_2(\bar{y} - y), \quad \mathbf{G}_y(y) = c_3,
\end{aligned} \tag{14}$$

with input current $u$ and constants $\hat{y}, \bar{y}, c_{1,2,3}$, whose exact values can be read from (Hodgkin & Huxley, 1952).

Thus, the multiscale ODEs (2) and the Hodgkin-Huxley equations are special case of the same general family (13) of ODEs. Moreover, the *gating functions* $\mathbf{G}_{y,z}(\mathbf{y})$, that model voltage-gated ion channels in the Hodgkin-Huxley equations, are similar in form to $\Delta \mathbf{t}_n, \overline{\Delta \mathbf{t}_n}$ in (2).

It is also worth highlighting the differences between our proposed model LEM (and the underlying ODE system (2)) and the Hodgkin-Huxley ODEs modeling the dynamics of the neuronal action potential. Given the complicated form of the nonlinearites $\mathbf{F}_{y,z}, \mathbf{G}_{y,z}, \mathbf{H}$ in the Hodgkin-Huxley equations (14), we cannot use them in designing any learning model. Instead, building on the abstract form of (13), we propose *bespoke* non-linearities in the ODE (2) to yield a tractable learning model, such as LEM (3). Moreover, it should be emphasized that the Hodgkin-Huxley equations only model the dynamics of a single neuron (with a scalar voltage and 3 ion channels), whereas the hidden state dimension $d$ of (2) can be arbitrary.

## D  RELATION BETWEEN LEM AND LSTM

The well-known LSTM (Hochreiter & Schmidhuber, 1997) (in its mainly-used version using a forget gate (Gers et al., 2000)) is given by,

$$\begin{aligned}
\mathbf{f}_n &= \hat{\sigma}(\mathbf{W}_f \mathbf{h}_{n-1} + \mathbf{V}_f \mathbf{u}_n + \mathbf{b}_f) \\
\mathbf{i}_n &= \hat{\sigma}(\mathbf{W}_i \mathbf{h}_{n-1} + \mathbf{V}_i \mathbf{u}_n + \mathbf{b}_i) \\
\mathbf{o}_n &= \hat{\sigma}(\mathbf{W}_o \mathbf{h}_{n-1} + \mathbf{V}_o \mathbf{u}_n + \mathbf{b}_o) \\
\mathbf{c}_n &= \mathbf{f}_n \odot \mathbf{c}_{n-1} + \mathbf{i}_n \odot \sigma(\mathbf{W} \mathbf{h}_{n-1} + \mathbf{V} \mathbf{u}_n + \mathbf{b}) \\
\mathbf{h}_n &= \mathbf{o}_n \odot \sigma(\mathbf{c}_n).
\end{aligned} \tag{15}$$

Here, for any $1 \leq n \leq N$, $\mathbf{h}_n \in \mathbb{R}^d$ is the hidden state and $\mathbf{c}_n \in \mathbb{R}^d$ is the so-called *cell state*. The vectors $\mathbf{i}_n, \mathbf{f}_n, \mathbf{o}_n \in \mathbb{R}^d$ are the *input, forget* and *output* gates, respectively. $\mathbf{u}_n \in \mathbb{R}^m$ is the input

signal and the weight matrices and bias vectors are given by $\mathbf{W}, \mathbf{W}_{f,i,o} \in \mathbb{R}^{d \times d}, \mathbf{V}, \mathbf{V}_{f,i,o} \in \mathbb{R}^{m \times d}$ and $\mathbf{b}, \mathbf{b}_{f,i,o} \in \mathbb{R}^d$, respectively.

It is straightforward to relate LSTM (15) and LEM (3) by first setting the cell state $\mathbf{c}_n = \mathbf{z}_n$, for all $1 \leq n \leq N$ and the hidden state $\mathbf{h}_n = \mathbf{y}_n$.

We further need to assume that the input state $\mathbf{i}_n = \mathbf{\Delta t}_n$ and the forget state has to be $\mathbf{f}_n = 1 - \mathbf{\Delta t}_n$. Finally, the output state of the LSTM (15) has to be

$$\mathbf{o}_n = \overline{\mathbf{\Delta t}}_n = 1, \quad \forall 1 \leq n \leq N.$$

Under these assumptions and by setting $\Delta t = 1$, we can readily observe that the LEM (3) and LSTM (15) are equivalent.

A different interpretation of LEM, in relation to LSTM, is as follows; LEM can be thought of a variant of LSTM but with two cell states $\mathbf{y}_n, \mathbf{z}_n$ per unit and no output gate. The input gates are $\mathbf{\Delta t}_n$ and $\overline{\mathbf{\Delta t}}_n$ and the forget gates are coupled to the input gates. Given that the state $\mathbf{z}_n$ is fed into the update for the state $\mathbf{y}_n$, one can think of one of the cell states sitting above the other, leading to a more sophisticated recursive update for LEM (3), when compared to LSTM (15).

Table 11: Test accuracies on EigenWorms using 5 re-trainings of each best performing network (based on the validation set) for LSTMs with $\Delta t$-scaled input and forget gates, as well as LSTMs with sub-sampling routines, baseline LSTM and LEM.

| Model | test accuracy | # units | # params |
|---|---|---|---|
| t-BPTT LSTM | $57.9\% \pm 7.0\%$ | 32 | 5.3k |
| sub-samp. LSTM | $69.2\% \pm 8.3\%$ | 32 | 5.3k |
| LSTM | $38.5\% \pm 10.1\%$ | 32 | 5.3k |
| $\Delta t$-LSTM v1 | $53.3\% \pm 8.2\%$ | 32 | 5.3k |
| $\Delta t$-LSTM v2 | $56.9\% \pm 6.7\%$ | 32 | 5.3k |
| LEM | $92.3\% \pm 1.8\%$ | 32 | 5.3k |

Another key difference between LEM and LSTM is the scaling of the learnable gates in LEM by the small hyperparameter $\Delta t$. It is natural to examine whether scaling LSTM with such a small hyperparameter $\Delta t$ will improve its performance on sequential tasks with long-term dependencies. To this end, we propose to *scale* the input and forget gate of an LSTM with a small hyperparameter in two different ways, where we denote the new forget gate as $\hat{\mathbf{f}}_n$ and the new input gate as $\hat{\mathbf{i}}_n$. The first version is ($\Delta t$-LSTM v1)

$$\hat{\mathbf{f}}_n = \Delta t \mathbf{f}_n, \quad \hat{\mathbf{i}}_n = \Delta t \mathbf{i}_n,$$

while the second version is ($\Delta t$-LSTM v2)

$$\hat{\mathbf{f}}_n = (1 - \Delta t) \mathbf{f}_n, \quad \hat{\mathbf{i}}_n = \Delta t \mathbf{i}_n.$$

We can see in Table 11 that both $\Delta t$-scaled versions of the LSTM lead to some improvements over the baseline LSTM for very long-sequence Eigenworms dataset, while still performing very poorly when compared to LEM. Moreover, we can see that standard sub-sampling routines, such as a truncation of the BPTT algorithm or random sub-sampling, applied to LSTMs lead to better improvements than $\Delta t$-scaling the forget and input gate.

# E  SUPPLEMENT TO THE RIGOROUS ANALYSIS OF LEM

In this section, we will provide detailed proofs of the propositions in Section 4 of the main article. We start with the following simplifying notation for various terms in LEM (3),

$$\mathbf{A}_{n-1} = \mathbf{W}_1 \mathbf{y}_{n-1} + \mathbf{V}_1 \mathbf{u}_n + \mathbf{b}_1,$$
$$\mathbf{B}_{n-1} = \mathbf{W}_2 \mathbf{y}_{n-1} + \mathbf{V}_2 \mathbf{u}_n + \mathbf{b}_2,$$
$$\mathbf{C}_{n-1} = \mathbf{W}_z \mathbf{y}_{n-1} + \mathbf{V}_z \mathbf{u}_n + \mathbf{b}_z,$$
$$\mathbf{D}_n = \mathbf{W}_y \mathbf{z}_n + \mathbf{V}_y \mathbf{u}_n + \mathbf{b}_y.$$

Note that for all $1 \leq n \leq N$, $\mathbf{A}_n, \mathbf{B}_n, \mathbf{C}_n, \mathbf{D}_n \in \mathbb{R}^d$. With the above notation, LEM (3) can be written componentwise, for each component $1 \leq i \leq d$ as,

$$
\begin{aligned}
\mathbf{z}_n^i &= \mathbf{z}_{n-1}^i + \Delta t \hat{\sigma}(\mathbf{A}_{n-1}^i)\sigma(\mathbf{C}_{n-1}^i) - \Delta t \hat{\sigma}(\mathbf{A}_{n-1}^i)\mathbf{z}_{n-1}^i, \\
\mathbf{y}_n^i &= \mathbf{y}_{n-1}^i + \Delta t \hat{\sigma}(\mathbf{B}_{n-1}^i)\sigma(\mathbf{D}_n^i) - \Delta t \hat{\sigma}(\mathbf{B}_{n-1}^i)\mathbf{y}_{n-1}^i.
\end{aligned}
\tag{16}
$$

Moreover, we will use the following *order*-notation,

$$
\begin{aligned}
\beta &= \mathcal{O}(\alpha), \text{for } \alpha, \beta \in \mathbb{R}_+ \quad \text{if there exists constants } \overline{C}, \underline{C} \text{ such that } \underline{C}\alpha \leq \beta \leq \overline{C}\alpha. \\
\mathbf{M} &= \mathcal{O}(\alpha), \text{for } \mathbf{M} \in \mathbb{R}^{d_1 \times d_2}, \alpha \in \mathbb{R}_+ \quad \text{if there exists constant } \overline{C} \text{ such that } \|\mathbf{M}\| \leq \overline{C}\alpha.
\end{aligned}
\tag{17}
$$

Note that the techniques of proof in this following three sub-sections burrow heavily from those introduced in Rusch & Mishra (2021a).

### E.1 PROOF OF PROPOSITION 4.1 OF MAIN TEXT.

First, we prove Proposition 4.1, which yields the bound (4) for the hidden states of LEM.

*Proof.* The proof of the bound (4) is split into 2 parts. We start with the first equation in (16) and rewrite it as,

$$
\mathbf{z}_n^i = \left(1 - \Delta t \hat{\sigma}(\mathbf{A}_{n-1}^i)\right)\mathbf{z}_{n-1}^i + \Delta t \hat{\sigma}(\mathbf{A}_{n-1}^i)\sigma(\mathbf{C}_{n-1}^i).
$$

Noting that the activation functions are such that $0 \leq \hat{\sigma}(x) \leq 1$, for all $x$ and $-1 \leq \sigma(x) \leq 1$, for all $x$ and using the fact that $\Delta t \leq 1$, we have from the above expression that,

$$
\begin{aligned}
\mathbf{z}_n^i &\leq \left(1 - \Delta t \hat{\sigma}(\mathbf{A}_{n-1}^i)\right)\max\left(\mathbf{z}_{n-1}^i, 1\right) + \Delta t \hat{\sigma}(\mathbf{A}_{n-1}^i)\max\left(\mathbf{z}_{n-1}^i, 1\right), \\
&\leq \max\left(\mathbf{z}_{n-1}^i, 1\right).
\end{aligned}
$$

By a symmetric argument, one can readily show that,

$$
\mathbf{z}_n^i \geq \min(-1, \mathbf{z}_{n-1}^i).
$$

Combining the above inequalities yields,

$$
\min(-1, \mathbf{z}_{n-1}^i) \leq \mathbf{z}_n^i \leq \max\left(\mathbf{z}_{n-1}^i, 1\right).
\tag{18}
$$

Iterating (18) over $n$ and using $z_0^i = 0$ for all $1 \leq i \leq d$ leads to,

$$
-1 \leq \mathbf{z}_n^i \leq 1, \quad \forall n, \quad \forall 1 \leq i \leq d.
\tag{19}
$$

An argument, identical to the derivation of (19), but for the hidden state $\mathbf{y}$ yields,

$$
-1 \leq \mathbf{y}_n^i \leq 1, \quad \forall n, \quad \forall 1 \leq i \leq d.
\tag{20}
$$

Thus, we have shown that the hidden states remain in the interval $[-1, 1]$, irrespective of the sequence length.

Next, we will use the following elementary identities in the proof,

$$
b(a - b) = \frac{a^2}{2} - \frac{b^2}{2} - \frac{1}{2}(a - b)^2,
\tag{21}
$$

for any $a, b \in \mathbb{R}$, and also,

$$
ab \leq \frac{\epsilon a^2}{2} + \frac{b^2}{2\epsilon}, \quad \forall \epsilon > 0.
\tag{22}
$$

We fix $1 \leq i \leq d$ and multiply the first equation in (16) with $\mathbf{z}_{n-1}^i$ and apply (21) to obtain,

$$\frac{(\mathbf{z}_n^i)^2}{2} = \frac{(\mathbf{z}_{n-1}^i)^2}{2} + \Delta t \hat{\sigma}(\mathbf{A}_{n-1}^i)\sigma(\mathbf{C}_{n-1}^i)\mathbf{z}_{n-1}^i - \Delta t \hat{\sigma}(\mathbf{A}_{n-1}^i)(\mathbf{z}_{n-1}^i)^2 + \frac{(\mathbf{z}_n^i - \mathbf{z}_{n-1}^i)^2}{2}$$

$$= \frac{(\mathbf{z}_{n-1}^i)^2}{2} + \Delta t \hat{\sigma}(\mathbf{A}_{n-1}^i)\sigma(\mathbf{C}_{n-1}^i)\mathbf{z}_{n-1}^i - \Delta t \hat{\sigma}(\mathbf{A}_{n-1}^i)(\mathbf{z}_{n-1}^i)^2$$

$$+ \frac{\Delta t^2}{2} \left( \hat{\sigma}(\mathbf{A}_{n-1}^i)\sigma(\mathbf{C}_{n-1}^i) - \hat{\sigma}(\mathbf{A}_{n-1}^i)\mathbf{z}_{n-1}^i \right)^2, \quad \text{(from (16))}$$

$$\leq \frac{(\mathbf{z}_{n-1}^i)^2}{2} + \Delta t \hat{\sigma}(\mathbf{A}_{n-1}^i)|\sigma(\mathbf{C}_{n-1}^i)||\mathbf{z}_{n-1}^i| - \Delta t \hat{\sigma}(\mathbf{A}_{n-1}^i)(\mathbf{z}_{n-1}^i)^2$$

$$+ \frac{\Delta t^2}{2}(\hat{\sigma}(\mathbf{A}_{n-1}^i)\sigma(\mathbf{C}_{n-1}^i))^2 + \frac{\Delta t^2}{2}\hat{\sigma}(\mathbf{A}_{n-1}^i)^2(\mathbf{z}_{n-1}^i)^2$$

$$+ \Delta t^2 \hat{\sigma}(\mathbf{A}_{n-1}^i)^2|\sigma(\mathbf{C}_{n-1}^i)||\mathbf{z}_{n-1}^i| \quad (\text{as } (a-b)^2 \leq a^2 + b^2 + 2|a||b|)$$

We fix $\epsilon = \frac{2-\Delta t}{1+\Delta t}$ in the elementary identity (22) to yield,

$$|\sigma(\mathbf{C}_{n-1}^i)||\mathbf{z}_{n-1}^i| \leq \frac{\sigma(\mathbf{C}_{n-1}^i)^2}{2\epsilon} + \frac{\epsilon(\mathbf{z}_{n-1}^i)^2}{2}$$

Applying this to the inequality for $(\mathbf{z}_n^i)^2$ leads to,

$$\frac{(\mathbf{z}_n^i)^2}{2} \leq \frac{(\mathbf{z}_{n-1}^i)^2}{2} + \left( \Delta t \hat{\sigma}(\mathbf{A}_{n-1}^i) + \Delta t^2 \hat{\sigma}(\mathbf{A}_{n-1}^i)^2 \right) \frac{\sigma(\mathbf{C}_{n-1}^i)^2}{2\epsilon}$$

$$- \Delta t \hat{\sigma}(\mathbf{A}_{n-1}^i) \left[ 1 - \frac{\epsilon}{2} - \frac{\Delta t \hat{\sigma}(\mathbf{A}_{n-1}^i)}{2} - \frac{\Delta t \hat{\sigma}(\mathbf{A}_{n-1}^i)\epsilon}{2} \right] (\mathbf{z}_{n-1}^i)^2.$$

Using the fact that $0 \leq \hat{\sigma}(x) \leq 1$ for all $x \in \mathbb{R}$, $\sigma^2 \leq 1$ and that $\epsilon = \frac{2-\Delta t}{1+\Delta t}$, we obtain from the last line of the previous equation that,

$$(\mathbf{z}_n^i)^2 \leq (\mathbf{z}_{n-1}^i)^2 + \frac{\Delta t + \Delta t^2}{\epsilon} \leq (\mathbf{z}_{n-1}^i)^2 + \frac{\Delta t(1 + \Delta t)^2}{2 - \Delta t}, \quad \forall 1 \leq n.$$

Iterating the above estimate over $n = 1, \ldots, \bar{n}$, for any $1 \leq \bar{n}$ and setting $\bar{n} = n$ yields,

$$(\mathbf{z}_n^i)^2 \leq (\mathbf{z}_0^i)^2 + n\frac{\Delta t(1 + \Delta t)^2}{2 - \Delta t},$$

$$\Rightarrow \quad (\mathbf{z}_n^i)^2 \leq t_n \frac{(1 + \Delta t)^2}{2 - \Delta t} \quad \text{as } \mathbf{z}_0^i = 0, \ t_n = n\Delta t.$$

Taking a square root in the above inequality yields,

$$|\mathbf{z}_n^i| \leq \overline{\Delta}\sqrt{t_n}, \quad \forall n, \quad \forall 1 \leq i \leq d. \tag{23}$$

with $\overline{\Delta}$ defined in the expression (4).

We can repeat the above argument with the hidden state $\mathbf{y}$ to obtain,

$$|\mathbf{y}_n^i| \leq \overline{\Delta}\sqrt{t_n}, \quad \forall n, \quad \forall 1 \leq i \leq d. \tag{24}$$

Combining (23) and (24) with the pointwise bounds (19) and (20) yields the desired bound (4). $\quad\square$

### E.2 PROOF OF PROPOSITION 4.2 OF MAIN TEXT.

*Proof.* We can apply the chain rule repeatedly (for instance as in Pascanu et al. (2013)) to obtain,

$$\frac{\partial \mathcal{E}_n}{\partial \theta} = \sum_{1 \leq k \leq n} \underbrace{\frac{\partial \mathcal{E}_n}{\partial \mathbf{X}_n} \frac{\partial \mathbf{X}_n}{\partial \mathbf{X}_k} \frac{\partial^+ \mathbf{X}_k}{\partial \theta}}_{\frac{\partial \mathcal{E}_n^{(k)}}{\partial \theta}}. \tag{25}$$

Here, the notation $\frac{\partial^+ \mathbf{X}_k}{\partial \theta}$ refers to taking the partial derivative of $\mathbf{X}_k$ with respect to the parameter $\theta$, while keeping the other arguments constant.

A straightforward application of the product rule yields,

$$\frac{\partial \mathbf{X}_n}{\partial \mathbf{X}_k} = \prod_{k < \ell \leq n} \frac{\partial \mathbf{X}_\ell}{\partial \mathbf{X}_{\ell-1}}. \tag{26}$$

For any $k < \ell \leq n$, a tedious yet straightforward computation yields the following representation formula,

$$\frac{\partial \mathbf{X}_\ell}{\partial \mathbf{X}_{\ell-1}} = \mathbf{I}_{2d \times 2d} + \Delta t \mathbf{E}^{\ell,\ell-1} + \Delta t^2 \mathbf{F}^{\ell,\ell-1}. \tag{27}$$

Here $\mathbf{E}^{\ell,\ell-1} \in \mathbb{R}^{2d \times 2d}$ is a matrix whose entries are given below. For any $1 \leq i \leq d$, we have,

$$\mathbf{E}_{2i-1,2j-1}^{\ell,\ell-1} \equiv 0, \quad j \neq i$$
$$\mathbf{E}_{2i-1,2i-1}^{\ell,\ell-1} = -\hat{\sigma}(\mathbf{A}_{\ell-1}^i),$$
$$\mathbf{E}_{2i-1,2j}^{\ell,\ell-1} = (\mathbf{W}_1)_{i,j} \hat{\sigma}'(\mathbf{A}_{\ell-1}^i) \left( \sigma(\mathbf{C}_{\ell-1}^i) - \mathbf{z}_{\ell-1}^i \right) + (\mathbf{W}_z)_{i,j} \hat{\sigma}(\mathbf{A}_{\ell-1}^i) \sigma'(\mathbf{C}_{\ell-1}^i), \ \forall 1 \leq j \leq d$$
$$\mathbf{E}_{2i,2j-1}^{\ell,\ell-1} = (\mathbf{W}_y)_{i,j} \hat{\sigma}(\mathbf{B}_{\ell-1}^i) \sigma'(\mathbf{D}_\ell^i), \ \forall 1 \leq j \leq d$$
$$\mathbf{E}_{2i,2j}^{\ell,\ell-1} = (\mathbf{W}_2)_{i,j} \hat{\sigma}'(\mathbf{B}_{\ell-1}^i) \left( \sigma(\mathbf{D}_\ell^i) - \mathbf{y}_{\ell-1}^i \right), \quad j \neq i$$
$$\mathbf{E}_{2i,2i}^{\ell,\ell-1} = -\hat{\sigma}(\mathbf{B}_{\ell-1}^i) + (\mathbf{W}_2)_{i,i} \hat{\sigma}'(\mathbf{B}_{\ell-1}^i) \left( \sigma(\mathbf{D}_\ell^i) - \mathbf{y}_{\ell-1}^i \right). \tag{28}$$

Similarly, $\mathbf{F}^{\ell,\ell-1} \in \mathbb{R}^{2d \times 2d}$ is a matrix whose entries are given below. For any $1 \leq i \leq d$, we have,

$$\mathbf{F}_{2i-1,j}^{\ell,\ell-1} \equiv 0, \quad \forall 1 \leq j \leq 2d,$$
$$\mathbf{F}_{2i,2j-1}^{\ell,\ell-1} = -(\mathbf{W}_y)_{i,j} \hat{\sigma}(\mathbf{A}_{\ell-1}^j) \hat{\sigma}(\mathbf{B}_{\ell-1}^i) \sigma'(\mathbf{D}_\ell^i), \quad 1 \leq j \leq d,$$
$$\mathbf{F}_{2i,2j}^{\ell,\ell-1} = \hat{\sigma}(\mathbf{B}_{\ell-1}^i) \sigma'(\mathbf{D}_\ell^i) \sum_{\lambda=1}^{d} (\mathbf{W}_y)_{i,\lambda} \left( \left( \sigma(\mathbf{C}_{\ell-1}^\lambda) - \mathbf{z}_{\ell-1}^\lambda \right) \hat{\sigma}'(\mathbf{A}_{\ell-1}^\lambda)(\mathbf{W}_1)_{\lambda,j} + \hat{\sigma}(\mathbf{A}_{\ell-1}^\lambda) \sigma'(\mathbf{C}_{\ell-1}^\lambda)(\mathbf{W}_z)_{\lambda,j} \right). \tag{29}$$

Using the fact that,

$$\sup_{x \in \mathbb{R}} \max \left\{ |\sigma(x)|, |\sigma'(x)|, |\hat{\sigma}(x)|, |\hat{\sigma}'(x)| \right\} \leq 1,$$

the pointwise bounds (4), the notation $t_n = n\Delta t$ for all $n$, the definition of $\eta$ (6) and the definition of matrix norms, we obtain that,

$$\|\mathbf{E}^{\ell,\ell-1}\|_\infty \leq \max \left\{ 1 + \|\mathbf{W}_z\|_\infty + (1 + \min(1, \overline{\Delta}\sqrt{t_\ell}))\|\mathbf{W}_1\|_\infty, 1 + \|\mathbf{W}_y\|_\infty + (1 + \min(1, \overline{\Delta}\sqrt{t_\ell}))\|\mathbf{W}_2\|_\infty \right\}$$
$$\leq 1 + (2 + \min(1, \overline{\Delta}\sqrt{t_\ell}))\eta. \tag{30}$$

By similar calculations, we obtain,

$$\|\mathbf{F}^{\ell,\ell-1}\|_\infty \leq \|\mathbf{W}_y\|_\infty \left( 1 + (1 + \min(1, \overline{\Delta}\sqrt{t_\ell}))\|\mathbf{W}_1\|_\infty + \|\mathbf{W}_z\|_\infty \right)$$
$$\leq \eta(1 + (2 + \min(1, \overline{\Delta}\sqrt{t_\ell}))\eta). \tag{31}$$

Applying (30) and (31) in the representation formula (27) and observing that $\Delta t \leq 1$ and $\ell \leq n$, we obtain.

$$\left\| \frac{\partial \mathbf{X}_\ell}{\partial \mathbf{X}_{\ell-1}} \right\|_\infty \leq 1 + \left( 1 + (2 + \min(1, \overline{\Delta}\sqrt{t_\ell}))\eta \right) \Delta t + \eta \left( 1 + (2 + \min(1, \overline{\Delta}\sqrt{t_\ell}))\eta \right) \Delta t^2,$$
$$\leq 1 + \frac{\Gamma}{2} \Delta t,$$

With

$$\Gamma = 2 (1 + \eta) (1 + 3\eta) \tag{32}$$

Using the expression (26) with the above inequality yields,

$$\left\| \frac{\partial \mathbf{X}_n}{\partial \mathbf{X}_k} \right\|_\infty \leq \left( 1 + \frac{\Gamma}{2} \Delta t \right)^{n-k}. \tag{33}$$

Next, we choose $\Delta t << 1$ small enough such that the following holds,

$$\left(1 + \frac{\Gamma}{2}\Delta t\right)^{n-k} \leq 1 + \Gamma(n-k)\Delta t, \tag{34}$$

for any $1 \leq k < n$.

Hence applying (34) in (33), we obtain,

$$\left\|\frac{\partial \mathbf{X}_n}{\partial \mathbf{X}_k}\right\|_\infty \leq 1 + \Gamma(n-k)\Delta t. \tag{35}$$

For the sake of definiteness, we fix any $1 \leq \alpha, \beta \leq d$ and set $\theta = (\mathbf{W}_y)_{\alpha,\beta}$ in the following. The following bounds for any other choice of $\theta \in \Theta$ can be derived analogously. Given this, it is straightforward to calculate from the structure of LEM (3) that entries of the vector $\frac{\partial^+ \mathbf{X}_k}{\partial(\mathbf{W}_y)_{\alpha,\beta}}$ are given by,

$$\begin{aligned}
\left(\frac{\partial^+ \mathbf{X}_k}{\partial(\mathbf{W}_y)_{\alpha,\beta}}\right)_j &\equiv 0, \quad \forall\, j \neq 2\alpha, \\
\left(\frac{\partial^+ \mathbf{X}_k}{\partial(\mathbf{W}_y)_{\alpha,\beta}}\right)_{2\alpha} &= \Delta t \hat{\sigma}(\mathbf{B}_{k-1}^\alpha)\sigma'(\mathbf{D}_k^\alpha)\mathbf{z}_k^\beta.
\end{aligned} \tag{36}$$

Hence, by the pointwise bounds (4), we obtain from (36) that

$$\left\|\frac{\partial^+ \mathbf{X}_k}{\partial(\mathbf{W}_y)_{\alpha,\beta}}\right\|_\infty \leq \Delta t \min(1, \overline{\Delta}\sqrt{t_k}). \tag{37}$$

Finally, it is straightforward to calculate from the loss function $\mathcal{E}_n = \frac{1}{2}\|\mathbf{y}_n - \overline{\mathbf{y}}_n\|^2$ that

$$\frac{\partial \mathcal{E}_n}{\partial \mathbf{X}_n} = \left[0, \mathbf{y}_n^1 - \bar{\mathbf{y}}^1, \ldots\ldots, 0, \mathbf{y}_n^d - \bar{\mathbf{y}}^d\right]. \tag{38}$$

Therefore, using the pointwise bounds (4) and the notation $\hat{\mathbf{Y}} = \|\bar{\mathbf{y}}\|_\infty$, we obtain

$$\left\|\frac{\partial \mathcal{E}_n}{\partial \mathbf{X}_n}\right\|_\infty \leq \hat{\mathbf{Y}} + \min(1, \overline{\Delta}\sqrt{t_n}). \tag{39}$$

Applying (35), (37) and (39) in the definition (25) yields,

$$\left|\frac{\partial \mathcal{E}_n^{(k)}}{\partial(\mathbf{W}_y)_{\alpha,\beta}}\right| \leq \Delta t \min(1, \overline{\Delta}\sqrt{t_k})\left(\hat{\mathbf{Y}} + \min(1, \overline{\Delta}\sqrt{t_n})\right)(1 + \Gamma(n-k)\Delta t). \tag{40}$$

Observing that $1 \leq k \leq n$, we see that $n - k \leq n$ and $t_k \leq t_n$. Therefore, (41) can be estimated for any $1 \leq k \leq n$ by,

$$\left|\frac{\partial \mathcal{E}_n^{(k)}}{\partial(\mathbf{W}_y)_{\alpha,\beta}}\right| \leq \Delta t \min(1, \overline{\Delta}\sqrt{t_n})\left(\hat{\mathbf{Y}} + \min(1, \overline{\Delta}\sqrt{t_n})\right)(1 + \Gamma t_n), \quad 1 \leq k \leq n. \tag{41}$$

Applying the bound (41) in (25) leads to the following bound on the total gradient,

$$\begin{aligned}
\left|\frac{\partial \mathcal{E}_n}{\partial(\mathbf{W}_y)_{\alpha,\beta}}\right| &\leq \sum_{k=1}^n \left|\frac{\partial \mathcal{E}_n^{(k)}}{\partial(\mathbf{W}_y)_{\alpha,\beta}}\right| \\
&\leq t_n \min(1, \overline{\Delta}\sqrt{t_n})\left(\hat{\mathbf{Y}} + \min(1, \overline{\Delta}\sqrt{t_n})\right)(1 + \Gamma t_n) \\
&\leq t_n(1 + \hat{\mathbf{Y}})(1 + \Gamma t_n) \\
&\leq (1 + \hat{\mathbf{Y}})t_n + (1 + \hat{\mathbf{Y}})\Gamma t_n^2
\end{aligned} \tag{42}$$

which is the desired bound (6) for $\theta = (\mathbf{W}_y)_{\alpha,\beta}$.

Moreover, for *long-term dependencies* i.e., $k << n$, we can set $t_k = k\Delta t < 1$, with $k$ independent of sequence length $n$, in (40) to obtain the following bound on the partial gradient,

$$\left| \frac{\partial \mathcal{E}_n^{(k)}}{\partial (\mathbf{W}_y)_{\alpha,\beta}} \right| \leq \Delta t^{\frac{3}{2}} \overline{\Delta} \sqrt{k} \left( 1 + \hat{\mathbf{Y}} \right) (1 + \Gamma t_n), \quad 1 \leq k << n. \tag{43}$$

$\square$

**Remark E.1.** *The bound* (6) *on the total gradient depends on* $t_n = n\Delta t$, *with* $n$ *being the sequence length and* $\Delta t \leq 1$, *a hyperparameter which can either be chosen a priori or determined through a hyperparameter tuning procedure. The proof of the bound* (42) *relies on* $\Delta t$ *being sufficiently small. It would be natural to choose* $\Delta t \sim n^{-s}$, *for some* $s \geq 0$. *Substituting this expression in* (6) *leads to a bound of the form,*

$$\left| \frac{\partial \mathcal{E}_n}{\partial \theta} \right| = \mathcal{O}\left( n^{2(1-s)} \right) \tag{44}$$

*If* $s = 1$, *then clearly* $\left| \frac{\partial \mathcal{E}_n}{\partial \theta} \right| = \mathcal{O}(1)$ *i.e., the total gradient is bounded. Clearly, the exploding gradient problem is mitigated in this case.*

*On the other hand, if* $s$ *takes another value, for instance* $s = \frac{1}{2}$ *which is empirically observed during the hyperparameter training (see Section B.1, then we can readily observe from* (44) *that* $\left| \frac{\partial \mathcal{E}_n}{\partial \theta} \right| = \mathcal{O}(n)$. *Thus in this case, the gradient can grow with sequence length* $n$ *but only linearly and not exponentially. Thus, the exploding gradient problem is also mitigated in this case.*

### E.3 Proof of Proposition 4.3 of Main Text.

To mitigate the vanishing gradient problem, we need to obtain a more precise characterization of the gradient $\frac{\partial \mathcal{E}_n^{(k)}}{\partial \theta}$ defined in (25). For the sake of definiteness, we fix any $1 \leq \alpha, \beta \leq d$ and set $\theta = (\mathbf{W}_y)_{\alpha,\beta}$ in the following. The following formulas for any other choice of $\theta \in \Theta$ can be derived analogously. Moreover, for simplicity of notation, we set the target function $\bar{\mathbf{X}}_n \equiv 0$.

Proposition 4.3 is a straightforward corollary of the following,

**Proposition E.2.** *Let* $\mathbf{y}_n, \mathbf{z}_n$ *be the hidden states generated by LEM* (3), *then we have the following representation formula for the hidden state gradient,*

$$\frac{\partial \mathcal{E}_n^{(k)}}{\partial \theta} = \Delta t \hat{\sigma}(\mathbf{B}_{k-1}^\alpha) \sigma'(\mathbf{D}_k^\alpha) \mathbf{z}_k^\beta \left( \mathbf{y}_n^\alpha - \bar{\mathbf{y}}_n^\alpha \right)$$

$$+ \Delta t^2 \hat{\sigma}(\mathbf{B}_{k-1}^\alpha) \sigma'(\mathbf{D}_k^\alpha) \mathbf{z}_k^\beta \left[ \sum_{j=1}^d \left( \mathbf{y}_n^j - \bar{\mathbf{y}}_n^j \right) \sum_{\ell=k+1}^n \hat{\sigma}'(\mathbf{B}_{\ell-1}^j) \left( \sigma(\mathbf{D}_\ell^j) - \mathbf{y}_{\ell-1}^j \right) (\mathbf{W}_2)_{j,2\alpha} \right] \tag{45}$$

$$+ \Delta t^2 \hat{\sigma}(\mathbf{B}_{k-1}^\alpha) \sigma'(\mathbf{D}_k^\alpha) \mathbf{z}_k^\beta \left[ \sum_{\ell=k+1}^n \hat{\sigma}(\mathbf{B}_{\ell-1}^\alpha) \left( \mathbf{y}_n^\alpha - \bar{\mathbf{y}}_n^\alpha \right) \right] + \mathcal{O}(\Delta t^3).$$

*Here, the constants in* $\mathcal{O}$ *could depend on* $\eta$ *defined in* (6) *(main text).*

*Proof.* The starting point for deriving an asymptotic formula for the hidden state gradient $\frac{\partial \mathcal{E}_n^{(k)}}{\partial \theta}$ is to observe from the representation formula (27), the bound (31) on matrices $\mathbf{F}^{\ell,\ell-1}$ and the order notation (17) that,

$$\frac{\partial \mathbf{X}_\ell}{\partial \mathbf{X}_{\ell-1}} = \mathbf{I}_{2d \times 2d} + \Delta t \mathbf{E}^{\ell,\ell-1} + \mathcal{O}(\Delta t^2), \tag{46}$$

as long as $\eta$ is independent of $\Delta t$.

By using induction and the bounds (30),(31), it is straightforward to calculate the following representation formula for the product,

$$\frac{\partial \mathbf{X}_n}{\partial \mathbf{X}_k} = \prod_{k < \ell \leq n} \frac{\partial \mathbf{X}_\ell}{\partial \mathbf{X}_{\ell-1}} = \mathbf{I}_{2d \times 2d} + \Delta t \sum_{\ell=k+1}^n \mathbf{E}^{\ell,\ell-1} + \mathcal{O}(\Delta t^2). \tag{47}$$

Recall that we have set $\theta = (\mathbf{W}_y)_{\alpha,\beta}$. Hence, by the expressions (38) and (36), a direct but tedious calculation leads to,

$$\frac{\partial \mathcal{E}_n}{\partial \mathbf{X}_n} \mathbf{I}_{2d \times 2d} \frac{\partial^+ \mathbf{X}_k}{\partial \theta} = \Delta t \hat{\sigma}(\mathbf{B}_{k-1}^\alpha) \sigma'(\mathbf{D}_k^\alpha) \mathbf{z}_k^\beta (\mathbf{y}_n^\alpha - \bar{\mathbf{y}}_n^\alpha), \tag{48}$$

$$\sum_{\ell=k+1}^{n} \frac{\partial \mathcal{E}_n}{\partial \mathbf{X}_n} \mathbf{E}^{\ell,\ell-1} \frac{\partial^+ \mathbf{X}_k}{\partial \theta} = \tag{49}$$

$$\Delta t \hat{\sigma}(\mathbf{B}_{k-1}^\alpha) \sigma'(\mathbf{D}_k^\alpha) \mathbf{z}_k^\beta \left[ \sum_{j=1}^{d} \left( \mathbf{y}_n^j - \bar{\mathbf{y}}_n^j \right) \sum_{\ell=k+1}^{n} \hat{\sigma}'(\mathbf{B}_{\ell-1}^j) \left( \sigma(\mathbf{D}_\ell^j) - \mathbf{y}_{\ell-1}^j \right) (\mathbf{W}_2)_{j,2\alpha} - \sum_{\ell=k+1}^{n} \hat{\sigma}(\mathbf{B}_{\ell-1}^\alpha) \left( \mathbf{y}_n^\alpha - \bar{\mathbf{y}}_n^\alpha \right) \right]. \tag{50}$$

Therefore, by substituting the above expression into the representation formula (47) yields the desired formula (45).

In order to prove the formula (7) (see Proposition 4.3 of main text), we focus our interest on long-term dependencies i.e., $k << n$. Then, a closer perusal of the expression in (48), together with the pointwise bounds (4) which implies that $\mathbf{y}_{k-1} \approx \mathcal{O}(\sqrt{\Delta t})$, results in the following,

$$\frac{\partial \mathcal{E}_n}{\partial \mathbf{X}_n} \mathbf{I}_{2d \times 2d} \frac{\partial^+ \mathbf{X}_k}{\partial \theta} = \mathcal{O}\left( \Delta t^{\frac{3}{2}} \right). \tag{51}$$

Similarly, we also obtain,

$$\Delta t \sum_{\ell=k+1}^{n} \frac{\partial \mathcal{E}_n}{\partial \mathbf{X}_n} \mathbf{E}^{\ell,\ell-1} \frac{\partial^+ \mathbf{X}_k}{\partial \theta} = \mathcal{O}\left( \Delta t^{\frac{3}{2}} \right). \tag{52}$$

Combining (51) and (52) results in the desired asymptotic bound (7). $\square$

**Remark E.3.** *The upper bound on the gradient* (6) *and the gradient asymptotic formula* (7) *impact the choice of the timestep hyperparameter* $\Delta t$. *For sequence length* $n$, *if we choose* $\Delta t \sim n^{-s}$, *with* $s \geq 0$, *we see from Remark E.1 that the upper bound on the total gradient scales like* $\mathcal{O}(n^{2(1-s)})$. *On the other hand, from* (7), *the gradient contribution from long-term dependencies will scale like* $\mathcal{O}(n^{\frac{-3s}{2}})$. *Hence, a small value of* $s \approx 0$, *will ensure that the gradient with respect to long-term dependencies will be* $\mathcal{O}(1)$. *However, the total gradient will behave like* $\mathcal{O}(n^2)$ *and possibly blow up fast. Similarly, setting* $s \approx 1$ *leads to a bounded gradient, while the contributions from long-term dependencies decay as fast as* $n^{\frac{-3}{2}}$. *Hence, one has to find a value of* $s$ *that balances both these requirements. Equilibrating them leads to* $s = \frac{4}{7}$, *ensuring that the total gradient grows sub-linearly while long-term dependencies still contribute with a sub-linear decay. This value is very close to the empirically observed value of* $s = \frac{1}{2}$ *which also ensures that the total gradient grows linearly and the contribution of long-term dependencies decays sub-linearly in the sequence length* $n$.

### E.4 PROOF OF PROPOSITION 4.4

*Proof.* To prove this proposition, we have to construct hidden states $\mathbf{y}_n, \mathbf{z}_n$, output state $\omega_n$, weight matrices $\mathbf{W}_{1,2,y,z}, \mathcal{W}_y, \mathbf{V}_{1,2,y,z}$ and bias vectors $\mathbf{b}_{1,2,y,z}$ such that LEM (3) with output state $\omega_n = \mathcal{W}_y \mathbf{y}_n$ approximates the dynamical system (8).

Let $R^* > R >> 1$ and $\epsilon^* < \epsilon$, be parameters to be defined later. By the theorem for universal approximation of continuous functions with neural networks with the tanh activation function $\sigma = \tanh$ (Barron, 1993), given $\epsilon^*$, there exist weight matrices $W_1 \in \mathbb{R}^{d_1 \times d_h}, V_1 \in \mathbb{R}^{d_1 \times d_u}, W_2 \in \mathbb{R}^{d_h \times d_1}$ and bias vector $b_1 \in \mathbb{R}^{d_1}$ such that the tanh neural network defined by,

$$\mathcal{N}_1(h, u) = W_2 \sigma \left( W_1 h + V_1 u + b_1 \right), \tag{53}$$

approximates the underlying function $\mathbf{f}$ in the following manner,

$$\max_{\max(\|h\|,\|u\|) < R^*} \|\mathbf{f}(h, u) - \mathcal{N}_1(h, u)\| \leq \epsilon^*. \tag{54}$$

Similarly, one can readily approximate the identity function $\mathbf{g}(h, u) = h$ with a tanh neural network of the form,

$$\bar{\mathcal{N}}_2(h) = \bar{W}_2 \sigma \left( \bar{W}_1 h \right), \tag{55}$$

such that

$$\max_{\|h\|, \|u\| < R^*} \|\mathbf{g}(h) - \mathcal{N}_2(h)\| \leq \epsilon^*. \tag{56}$$

Next, we define the following dynamical system,

$$\begin{aligned}
\bar{\mathbf{z}}_n &= W_2 \sigma \left( W_1 \bar{\mathbf{y}}_{n-1} + V_1 \mathbf{u}_n + b_1 \right), \\
\bar{\mathbf{y}}_n &= \bar{W}_2 \sigma \left( \bar{W}_1 \bar{\mathbf{z}}_n \right),
\end{aligned} \tag{57}$$

with initial states $\bar{\mathbf{z}}_0 = \bar{\mathbf{y}}_0 = 0$.

Using the approximation bound (54), we derive the following bound,

$$\begin{aligned}
\|\phi_n - \bar{\mathbf{y}}_n\| &= \left\| \mathbf{f} \left( \phi_{n-1}, \mathbf{u}_n \right) - \bar{\mathbf{z}}_n + \bar{\mathbf{z}}_n - \bar{\mathbf{y}}_n \right\| \\
&\leq \left\| \mathbf{f} \left( \phi_{n-1}, \mathbf{u}_n \right) - W_2 \sigma \left( W_1 \bar{\mathbf{y}}_{n-1} + V_1 \mathbf{u}_n + b_1 \right) \right\| + \left\| \mathbf{g}(\bar{\mathbf{z}}_n) - \bar{W}_2 \sigma \left( \bar{W}_1 \bar{\mathbf{z}}_n \right) \right\| \\
&\leq \left\| \mathbf{f} \left( \phi_{n-1}, \mathbf{u}_n \right) - \mathbf{f} \left( \bar{\mathbf{y}}_{n-1}, \mathbf{u}_n \right) \right\| + \left\| \mathbf{f} \left( \bar{\mathbf{y}}_{n-1}, \mathbf{u}_n \right) - W_2 \sigma \left( W_1 \bar{\mathbf{y}}_{n-1} + V_1 \mathbf{u}_n + b_1 \right) \right\| \\
&\quad + \left\| \mathbf{g}(\bar{\mathbf{z}}_n) - \bar{W}_2 \sigma \left( \bar{W}_1 \bar{\mathbf{z}}_n \right) \right\| \\
&\leq \mathrm{Lip}(\mathbf{f}) \|\phi_{n-1} - \bar{\mathbf{y}}_{n-1}\| + 2\epsilon^* \quad \text{(from (54), (56))}.
\end{aligned}$$

Here, $\mathrm{Lip}(\mathbf{f})$ is the Lipschitz constant of the function $\mathbf{f}$ on the compact set $\{(h, u) \in \mathbb{R}^{d_h \times d_u} : \|h\|, \|u\| < \mathbb{R}^*\}$. Note that one can readily prove using the fact that $\bar{\mathbf{y}}_0 = \bar{\mathbf{z}}_0 = 0$, bounds (54), (56) and the assumption $\|\phi_n\|, \|\mathbf{u}_n\| < R$, that $\|\bar{\mathbf{z}}_n\|, \|\bar{\mathbf{y}}_n\| < R^* = 2R$.

Iterating the above inequality over $n$ leads to the bound,

$$\|\phi_n - \bar{\mathbf{y}}_n\| \leq 2\epsilon^* \sum_{\lambda=0}^{n-1} \mathrm{Lip}(\mathbf{f})^\lambda. \tag{58}$$

Hence, using the Lipschitz continuity of the output function $\mathbf{o}$ in (8), one obtains,

$$\|\mathbf{o}_n - \mathbf{o}(\bar{\mathbf{y}}_n)\| \leq 2\epsilon^* \mathrm{Lip}(\mathbf{o}) \sum_{\lambda=0}^{n-1} \mathrm{Lip}(\mathbf{f})^\lambda, \tag{59}$$

with $\mathrm{Lip}(\mathbf{o})$ being the Lipschitz constant of the function $\mathbf{o}$ on the compact set $\{h \in \mathbb{R}^{d_h} : \|h\| < R^*\}$.

Next, we can use the universal approximation theorem for neural networks again to conclude that given a tolerance $\bar{\epsilon}$, there exist weight matrices $W_3 \in \mathbb{R}^{d_2 \times d_h}, W_4 \in \mathbb{R}^{d_h \times d_2}$ and bias vector $b_2 \in \mathbb{R}^{d_2}$ such that the tanh neural network defined by,

$$\mathcal{N}_3(h) = W_4 \sigma \left( W_3 h + b_2 \right), \tag{60}$$

approximates the underlying output function $\mathbf{o}$ in the following manner,

$$\max_{\|h\| < R^*} \|\mathbf{o}(h) - \mathcal{N}_3(h)\| \leq \bar{\epsilon}. \tag{61}$$

Now defining,

$$\bar{\omega}_n = W_4 \sigma \left( W_3 \bar{\mathbf{y}}_n + b_2 \right), \tag{62}$$

we obtain from (61) and (59) that,

$$\|\mathbf{o}_n - \bar{\omega}_n\| \leq \bar{\epsilon} + 2\epsilon^* \mathrm{Lip}(\mathbf{o}) \sum_{\lambda=0}^{n-1} \mathrm{Lip}(\mathbf{f})^\lambda. \tag{63}$$

Next, we introduce the notation,

$$\tilde{\mathbf{z}}_n = \sigma \left( W_1 \bar{\mathbf{y}}_{n-1} + V_1 \mathbf{u}_n + b_1 \right), \quad \tilde{\mathbf{y}}_n = \sigma \left( \bar{W}_1 \bar{\mathbf{z}}_n \right). \tag{64}$$

From (57), we see that

$$\bar{\mathbf{z}}_n = W_2 \tilde{\mathbf{z}}_n, \quad \bar{\mathbf{y}}_n = \bar{W}_2 \tilde{\mathbf{y}}_n \tag{65}$$

Thus from (65) and (63), we have

$$
\begin{aligned}
\bar{\omega}_n &= W_4 \sigma \left( W_3 W_2 \tilde{\mathbf{y}}_n + b_2 \right), \\
&= W_4 \sigma \left( W_3 W_2 \sigma \left( \bar{W}_1 W_2 \tilde{\mathbf{z}}_n \right) + b_2 \right).
\end{aligned}
\tag{66}
$$

Define the function $\mathcal{R} : \mathbb{R}^{d_h} \times \mathbb{R}^{d_u} \mapsto \mathbb{R}^{d_o}$ by,

$$
\mathcal{R}(z) = W_4 \sigma \left( W_3 W_2 \sigma \left( \bar{W}_1 W_2 z \right) + b_2 \right).
\tag{67}
$$

The function, defined above, is clearly Lipschitz continuous. We can apply the universal approximation theorem for tanh neural networks to find, for any given tolerance $\tilde{\epsilon}$, weight matrices $W_5 \in \mathbb{R}^{d_3 \times d_4}, W_6 \in \mathbb{R}^{d_o \times d_3}, V_2 \in \mathbb{R}^{d_3 \times d_u}$ and bias vector $b_3 \in \mathbb{R}^{d_3}$ such that the following holds,

$$
\max_{\max(\|z\|) < R^*} \| \mathcal{R}(z) - W_6 \sigma(W_5 z + b_3) \| \leq \tilde{\epsilon}.
\tag{68}
$$

Denote $\tilde{\omega}_n = W_6 \sigma(W_5 \tilde{\mathbf{z}}_n + b_3)$, then from (68) and (66), we obtain that

$$
\| \bar{\omega}_n - \tilde{\omega}_n \| \leq \tilde{\epsilon}.
$$

Combining this estimate with (63) yields,

$$
\| \mathbf{o}_n - \tilde{\omega}_n \| \leq \tilde{\epsilon} + \bar{\epsilon} + 2\epsilon^* \mathrm{Lip}(\mathbf{o}) \sum_{\lambda=0}^{n-1} \mathrm{Lip}(\mathbf{f})^{\lambda}.
\tag{69}
$$

Now, we collect all ingredients to define the LEM that can approximate the dynamical system (8). To this end, we define hidden states $\mathbf{z}_n, \mathbf{y}_n \in \mathbb{R}^{2d_h}$ as

$$
\mathbf{z}_n = [\tilde{\mathbf{z}}_n, \hat{\mathbf{z}}_n], \quad \mathbf{y}_n = [\tilde{\mathbf{y}}_n, \hat{\mathbf{y}}_n],
$$

with $\tilde{\mathbf{z}}_n, \hat{\mathbf{z}}_n, \tilde{\mathbf{y}}_n, \hat{\mathbf{y}}_n \in \mathbb{R}^{d_h}$. These hidden states are evolved by the dynamical system,

$$
\begin{aligned}
\mathbf{z}_n^\perp &= \sigma \left( \begin{bmatrix} W_1 \bar{W}_2 & 0 \\ 0 & 0 \end{bmatrix} \mathbf{y}_{n-1}^\perp + [V_1 \mathbf{u}_n, 0]^\perp + [b_1, 0]^\perp \right), \\
\mathbf{y}_n^\perp &= \sigma \left( \begin{bmatrix} \bar{W}_1 W_2 & 0 \\ W_5 & 0 \end{bmatrix} \mathbf{z}_n^\perp + [0, 0]^\perp + [0, b_3]^\perp \right)
\end{aligned}
\tag{70}
$$

and the output state is calculated by,

$$
\omega_n^\perp = [0, W_6] \mathbf{y}_n^\perp.
\tag{71}
$$

Finally, we can recast the dynamical system (70), (71) as a LEM of the form (3) for the hidden states $\mathbf{y}_n, \mathbf{z}_n$, defined in (70), with the following parameters, Now, define the hidden states $\bar{\mathbf{y}}_n, \bar{\mathbf{z}}_n \in \mathbb{R}^{d_y}$ for all $1 \leq n \leq N$ by the LEM (3) with the following parameters,

$$
\begin{aligned}
\Delta t &= 1, \quad d_y = 2d_h, \\
\mathbf{W}_1 &= \mathbf{W}_2 = \mathbf{V}_1 = \mathbf{V}_2 = 0 \\
\mathbf{b}_1 &= \mathbf{b}_2 = \mathbf{b}_\infty, \\
\mathbf{W}_z &= \begin{bmatrix} W_1 \bar{W}_2 & 0 \\ 0 & 0 \end{bmatrix}, \quad \mathbf{V}_z = [V_1, 0], \quad \mathbf{b}_z = [b_1, 0] \\
\mathbf{W}_y &= \begin{bmatrix} \bar{W}_1 W_2 & 0 \\ W_5 & 0 \end{bmatrix}, \quad \mathbf{V}_y = 0, \quad \mathbf{b}_z = [0, b_3]. \\
\mathcal{W}_y &= [0, W_6].
\end{aligned}
\tag{72}
$$

Here, $\mathbf{b}_\infty \in \mathbb{R}^{d_h}$ is defined as

$$
\mathbf{b}_\infty = [b_\infty, b_\infty, \ldots, \ldots, b_\infty],
$$

with $1 << b_\infty$ is such that

$$
|1 - \hat{\sigma}(b_\infty)| \leq \delta.
\tag{73}
$$

The nature of the sigmoid function guarantees the existence of such a $b_\infty$ for any $\delta$. As $\delta$ decays exponentially fast, we set it to 0 in the following for notational simplicity.

It is straightforward to verify that the output state of the LEM (3) with parameters given in (72) is $\omega_n = \tilde{\omega}_n$.

Therefore, from (69) and by setting $\bar{\epsilon} < \frac{\epsilon}{3}$, $\tilde{\epsilon} < \frac{\epsilon}{3}$ and

$$\epsilon^* < \frac{\epsilon}{6\text{Lip}(\mathbf{o}) \sum\limits_{\lambda=0}^{N-1} \text{Lip}(\mathbf{f})^\lambda},$$

we prove the desired bound (9).

$\square$

### E.5 PROOF OF PROPOSITION 4.5

*Proof.* The proof of this proposition is based heavily on the proof of Proposition 4.4. Hence, we will highlight the main points of difference.

As the steps for approximation of a general Lipschitz continuous output map are identical to the corresponding steps in the proof of proposition 4.4 (see the steps from Eqns. (59) to (69)), we will only consider the following linear output map for convenience herein,

$$\mathbf{o}(\boldsymbol{\psi}_n) = \mathcal{W}_c \boldsymbol{\psi}_n. \tag{74}$$

Let $R^* > R >> 1$ and $\epsilon^* < \epsilon$, be parameters to be defined later. By the theorem for universal approximation of continuous functions with neural networks with the tanh activation function $\sigma = \tanh$, given $\epsilon^*$, there exist weight matrices $W_1^f, W_2^f \in \mathbb{R}^{d_1 \times d_h}, V_1^f \in \mathbb{R}^{d_1 \times d_u}, W_3^f \in \mathbb{R}^{d_h \times d_1}$ and bias vector $b_1^f \in \mathbb{R}^{d_1}$ such that the tanh neural network defined by,

$$\mathcal{N}_f(h, c, u) = W_3^f \sigma \left( W_1^f h + W_2^f c + V_1^f u + b_1^f \right), \tag{75}$$

approximates the underlying function $\mathbf{f}$ in the following manner,

$$\max_{\max(\|h\|, \|c\|, \|u\|) < R^*} \|\mathbf{f}(h, c, u) - \mathcal{N}_f(h, c, u)\| \leq \epsilon^*. \tag{76}$$

Next, we define the following map,

$$\mathbf{G}(h, c, u) = \mathbf{g}(h, c, u) + \left( 1 - \frac{1}{\tau} \right) c, \tag{77}$$

for any $\tau > 0$.

By the universal approximation theorem, given $\epsilon^*$, there exist weight matrices $W_1^g, W_2^g \in \mathbb{R}^{d_2 \times d_h}, V_1^g \in \mathbb{R}^{d_2 \times d_u}, W_3^g \in \mathbb{R}^{d_h \times d_2}$ and bias vector $b_1^g \in \mathbb{R}^{d_2}$ such that the tanh neural network defined by,

$$\mathcal{N}_g(h, c, u) = W_3^g \sigma \left( W_1^g h + W_2^g c + V_1^g u + b_1^g \right), \tag{78}$$

approximates the function $\mathbf{G}$ (77) in the following manner,

$$\max_{\max(\|h\|, \|c\|, \|u\|) < R^*} \|\mathbf{G}(h, c, u) - \mathcal{N}_f(h, c, u)\| \leq \epsilon^*. \tag{79}$$

Note that the sizes of the neural network $\mathcal{N}_g$ can be made independent of the small parameter $\tau$ by simply taking the sum of the neural networks approximating the functions $g$ and $\hat{g}(h, c, u) = c$ with tanh neural networks. As neither of these functions depend on the small parameter $\tau$, the sizes of the corresponding neural networks are independent of the small parameter too.

Next, as in the proof of proposition 4.4, one can readily approximate the identity function $\hat{f}(h, c, u) = h$ with a tanh neural network of the form,

$$\bar{\mathcal{N}}_f(h) = \bar{W}_2 \sigma \left( \bar{W}_1 h \right), \tag{80}$$

such that

$$\max_{\|h\|, \|c\|, \|u\| < R^*} \|\hat{f}(h, c, u) - \mathcal{N}_f(h)\| \leq \epsilon^*, \tag{81}$$

and with the same weights and biases, one can approximate the identity function $\hat{g}(h, c, u) = c$ with the tanh neural network,

$$\bar{\mathcal{N}}_g(c) = \bar{W}_2 \sigma \left( \bar{W}_1 c \right), \tag{82}$$

such that

$$\max_{\|h\|, \|c\|, \|u\| < R^*} \|\hat{g}(h, c, u) - \mathcal{N}_g(c)\| \leq \epsilon^*. \tag{83}$$

Next, we define the following dynamical system,

$$
\begin{aligned}
\hat{\mathbf{z}}_n &= W_3^f \sigma \left( W_1^f \tilde{\mathbf{y}}_{n-1} + W_2^f \hat{\mathbf{y}}_{n-1} + V_1^f \mathbf{u}_n + b_1^f \right), \\
\tilde{\mathbf{z}}_n &= \bar{W}_2 \sigma \left( \bar{W}_1 \hat{\mathbf{y}}_{n-1} \right), \\
\hat{\mathbf{y}}_n &= (1 - \tau) \hat{\mathbf{y}}_{n-1} + \tau W_3^g \sigma \left( W_1^g \hat{\mathbf{z}}_n + W_2^g \tilde{\mathbf{z}}_n + V_1^g \mathbf{u}_n + b_1^g \right), \\
\tilde{\mathbf{y}}_n &= \bar{W}_2 \sigma \left( \bar{W}_1 \hat{\mathbf{z}}_n \right),
\end{aligned}
\tag{84}
$$

with hidden states $\hat{\mathbf{z}}_n, \tilde{\mathbf{z}}_n, \hat{\mathbf{y}}_n, \tilde{\mathbf{y}}_n \in \mathbb{R}^{d_h}$ and with initial states $\hat{\mathbf{z}}_0 = \tilde{\mathbf{z}}_0 = \hat{\mathbf{y}}_0 = \tilde{\mathbf{y}}_0 = 0$.

We derive the following bounds,

$$
\begin{aligned}
\|\boldsymbol{\phi}_n - \hat{\mathbf{z}}_n\| &= \|\mathbf{f}(\boldsymbol{\phi}_{n-1}, \boldsymbol{\psi}_{n-1}, \mathbf{u}_n) - W_3^f \sigma \left( W_1^f \tilde{\mathbf{y}}_{n-1} + W_2^f \hat{\mathbf{y}}_{n-1} + V_1^f \mathbf{u}_n + b_1^f \right) \| \\
&\leq \|\mathbf{f}(\boldsymbol{\phi}_{n-1}, \boldsymbol{\psi}_{n-1}, \mathbf{u}_n) - \mathbf{f}(\tilde{\mathbf{y}}_{n-1}, \hat{\mathbf{z}}_{n-1}, \mathbf{u}_n)\|, \\
&\quad + \|\mathbf{f}(\tilde{\mathbf{y}}_{n-1}, \hat{\mathbf{z}}_{n-1}, \mathbf{u}_n) - W_3^f \sigma \left( W_1^f \tilde{\mathbf{y}}_{n-1} + W_2^f \hat{\mathbf{y}}_{n-1} + V_1^f \mathbf{u}_n + b_1^f \right) \| \\
&\leq \mathrm{Lip}(\mathbf{f}) \left( \|\boldsymbol{\phi}_{n-1} - \hat{\mathbf{z}}_{n-1}\| + 2\|\tilde{\mathbf{y}}_{n-1} - \hat{\mathbf{z}}_{n-1}\| + \|\boldsymbol{\psi}_{n-1} - \tilde{\mathbf{y}}_{n-1}\| \right) + \epsilon^* \quad \text{(by (79))} \\
&\leq \mathrm{Lip}(\mathbf{f}) \left( \|\boldsymbol{\phi}_{n-1} - \hat{\mathbf{z}}_{n-1}\| + \|\boldsymbol{\psi}_{n-1} - \tilde{\mathbf{y}}_{n-1}\| \right) + (1 + 2\mathrm{Lip}(\mathbf{f})) \epsilon^* \quad \text{(by (81), (84))},
\end{aligned}
$$

and

$$
\begin{aligned}
\|\boldsymbol{\psi}_n - \hat{\mathbf{y}}_n\| &= \|(1 - \tau)(\boldsymbol{\psi}_{n-1} - \hat{\mathbf{y}}_{n-1}) + \tau \left( \mathbf{G}(\boldsymbol{\phi}_n, \boldsymbol{\psi}_{n-1}, \mathbf{u}_n) - W_3^g \sigma \left( W_2^g \tilde{\mathbf{z}}_n + W_1^g \hat{\mathbf{z}}_n + V_1^g \mathbf{u}_n + b_1^g \right) \right) \| \\
&\leq \|\boldsymbol{\psi}_{n-1} - \hat{\mathbf{y}}_{n-1}\| + \tau \|\mathbf{G}(\boldsymbol{\phi}_n, \boldsymbol{\psi}_{n-1}, \mathbf{u}_n) - \mathbf{G}(\hat{\mathbf{z}}_n, \tilde{\mathbf{z}}_n, \mathbf{u}_n)\| \\
&\quad + \tau \|\mathbf{G}(\hat{\mathbf{z}}_n, \tilde{\mathbf{z}}_n, \mathbf{u}_n) - W_3^g \sigma \left( W_2^g \tilde{\mathbf{z}}_n + W_1^g \hat{\mathbf{z}}_n + V_1^g \mathbf{u}_n + b_1^g \right) \| \\
&\leq \|\boldsymbol{\psi}_{n-1} - \hat{\mathbf{y}}_{n-1}\| + \tau \mathrm{Lip}(\mathbf{G}) \left( \|\boldsymbol{\phi}_n - \hat{\mathbf{z}}_n\| + \|\tilde{\mathbf{z}}_n - \hat{\mathbf{y}}_{n-1}\| + \|\boldsymbol{\psi}_{n-1} - \hat{\mathbf{y}}_{n-1}\| \right) + \tau \epsilon^*, \\
&\leq (1 + \tau \mathrm{Lip}(\mathbf{G}))(1 + \mathrm{Lip}(\mathbf{f}))\|\boldsymbol{\psi}_{n-1} - \hat{\mathbf{y}}_{n-1}\| + \tau \mathrm{Lip}(\mathbf{G})\mathrm{Lip}(\mathbf{f})\|\boldsymbol{\phi}_{n-1} - \hat{\mathbf{z}}_{n-1}\| \\
&\quad + \tau (1 + \mathrm{Lip}(\mathbf{G})(2 + 2\mathrm{Lip}(\mathbf{f})))\epsilon^*,
\end{aligned}
$$

where the last inequality follows by using the previous inequality together with (84) and (83).

As $\tau < 1$, it is easy to see from (77) that $\mathrm{Lip}(\mathbf{G}) < \mathrm{Lip}(\mathbf{g}) + \frac{2}{\tau}$. Therefore, the last inequality reduces to,

$$
\begin{aligned}
\|\boldsymbol{\psi}_n - \hat{\mathbf{y}}_n\| &\leq (3 + \tau \mathrm{Lip}(\mathbf{g}))(1 + \mathrm{Lip}(\mathbf{f}))\|\boldsymbol{\psi}_{n-1} - \hat{\mathbf{y}}_{n-1}\| + (2 + \tau \mathrm{Lip}(\mathbf{g}))\mathrm{Lip}(\mathbf{f})\|\boldsymbol{\phi}_{n-1} - \hat{\mathbf{z}}_{n-1}\| \\
&\quad + (\tau + (2 + \tau \mathrm{Lip}(\mathbf{g}))(2 + 2\mathrm{Lip}(\mathbf{f})))\epsilon^*.
\end{aligned}
$$

Adding we obtain,

$$\|\boldsymbol{\phi}_n - \hat{\mathbf{z}}_n\| + \|\boldsymbol{\psi}_n - \hat{\mathbf{y}}_n\| \leq C^* \left( \|\boldsymbol{\phi}_{n-1} - \hat{\mathbf{z}}_{n-1}\| + \|\boldsymbol{\psi}_{n-1} - \hat{\mathbf{y}}_{n-1}\| \right) + D^* \epsilon^*, \tag{85}$$

where,

$$
\begin{aligned}
C^* &= \max\{(3 + \mathrm{Lip}(\mathbf{g}))\mathrm{Lip}(\mathbf{f}), \mathrm{Lip}(\mathbf{f})(3 + \mathrm{Lip}(\mathbf{g}))(1 + \mathrm{Lip}(\mathbf{f}))\}, \\
D^* &= 1 + (2 + \mathrm{Lip}(\mathbf{g}))(2 + 2\mathrm{Lip}(\mathbf{f})).
\end{aligned}
\tag{86}
$$

Iterating over $n$ leads to the bound,

$$\|\boldsymbol{\phi}_n - \hat{\mathbf{z}}_n\| + \|\boldsymbol{\psi}_n - \hat{\mathbf{y}}_n\| \leq \epsilon^* D^* \sum_{\lambda=0}^{n-1} (C^*)^\lambda. \tag{87}$$

Here, $\mathrm{Lip}(\mathbf{f}), \mathrm{Lip}(\mathbf{g})$ are the Lipschitz constants of the functions $\mathbf{f}, \mathbf{g}$ on the compact set $\{(h, c, u) \in \mathbb{R}^{d_h \times d_h \times d_u} : \|h\|, \|c\|, \|u\| < \mathbb{R}^*\}$. Note that one can readily prove using the zero values of initial states, the bounds (81), (83) and the assumption $\|\boldsymbol{\phi}_n\|, \|\boldsymbol{\psi}_n\|, \|\mathbf{u}_n\| < R$, that $\|\hat{\mathbf{z}}_n\|, \|\tilde{\mathbf{z}}_n\|, \|\hat{\mathbf{y}}_n\|, \|\tilde{\mathbf{y}}_n\| < R^* = 2R$.

Using the definition of the output function (11) and the bound (87) that,

$$\|\mathbf{o}_n - \mathbf{o}(\hat{\mathbf{y}}_n)\| \leq \|\mathcal{W}_c\| \epsilon^* D^* \sum_{\lambda=0}^{n-1} (C^*)^{\lambda}. \tag{88}$$

Defining the dynamical system,

$$\begin{aligned}
\mathbf{z}_n^* &= \sigma \left( W_1^f \bar{W}_2 \bar{\mathbf{y}}_{n-1} + W_2^f W_3^g \mathbf{y}_{n-1}^* + V_1^f \mathbf{u}_n + b_1^f \right) \\
\bar{\mathbf{z}}_n &= \sigma \left( \bar{W}_1 W_3^g \mathbf{y}_{n-1}^* \right) \\
\mathbf{y}_n^* &= (1-\tau) \mathbf{y}_{n-1}^* + \tau \sigma \left( W_1^g W_3^f \mathbf{z}_n^* + W_2^g \bar{W}_2 \bar{\mathbf{z}}_n + V_1^g \mathbf{u}_n + b_1^g \right), \\
\bar{\mathbf{y}}_n &= \sigma \left( \bar{W}_1 W_3^f \mathbf{z}_n^* \right).
\end{aligned} \tag{89}$$

By multiplying suitable matrices to (84), we obtain that,

$$\hat{\mathbf{z}}_n = W_3^f \mathbf{z}_n^*, \quad \tilde{\mathbf{z}}_n = \bar{W}_2 \bar{\mathbf{z}}_n, \quad \hat{\mathbf{y}}_n = W_3^g \mathbf{y}_n^*, \quad \tilde{\mathbf{y}}_n = \bar{W}_2 \bar{\mathbf{y}}_n. \tag{90}$$

Finally, in addition to $b_\infty$ defined in (58), for any given $\tau \in (0, 1]$, we introduce $b_\tau \in \mathbb{R}$ defined by

$$\hat{\sigma}(b_\tau) = \tau. \tag{91}$$

The existence of a unique $b_\tau$ follows from the fact that the sigmoid function $\hat{\sigma}$ is monotone. Next, we define the two vectors $\mathbf{b}_\infty, \mathbf{b}_\tau \in \mathbb{R}^{2d_h}$ as

$$\begin{aligned}
\mathbf{b}_\infty^i &= b_\infty, \quad \forall\, 1 \leq i \leq 2d_h, \\
\mathbf{b}_\tau^i &= b_\tau, \quad \forall\, 1 \leq i \leq d_h, \\
\mathbf{b}_\tau^i &= b_\infty, \quad \forall\, d_h + 1 \leq i \leq 2d_h.
\end{aligned} \tag{92}$$

We are now in a position to define the LEM of form (3), which will approximate the two-scale dynamical system (10). To this end, we define the hidden states $\mathbf{z}_n, \mathbf{y}_n \in \mathbb{R}^{2d_h}$ such that $\mathbf{z}_n = [\mathbf{z}_n^*, \bar{\mathbf{z}}_n]$ and $\mathbf{y}_n = [\mathbf{y}_n^*, \bar{\mathbf{y}}_n]$. The parameters for the corresponding LEM of form (3) given by,

$$\begin{aligned}
&\Delta t = 1, d_y = 2d_h \\
&\mathbf{W}_1 = \mathbf{W}_2 = \mathbf{V}_1 = \mathbf{V}_2 \equiv 0, \\
&\mathbf{b}_1 = \mathbf{b}_\infty, \quad \mathbf{b}_2 = \mathbf{b}_\tau, \\
&\mathbf{W}_z = \begin{bmatrix} W_2^f W_3^g & W_1^f \bar{W}_2 \\ \bar{W}_1 W_3^g & 0 \end{bmatrix}, \quad \mathbf{V}_z = [V_1^f 0], \quad \mathbf{b}_z = [b_1^f, 0], \\
&\mathbf{W}_y = \begin{bmatrix} W_1^g W_3^f & W_2^g \bar{W}_2 \\ \bar{W}_1 W_3^f & 0 \end{bmatrix}, \quad \mathbf{V}_z = [V_1^g 0], \quad \mathbf{b}_z = [b_1^g, 0],
\end{aligned} \tag{93}$$

and with following parameters defining the output states,

$$\mathcal{W}_y = [\mathcal{W}_c W_3^g \; 0], \tag{94}$$

yields an output state $\omega_n = \mathcal{W}_y \mathbf{y}_n$.

It is straightforward to observe that $\omega_n \equiv \mathbf{o}(\hat{\mathbf{y}}_n)$. Hence, the desired bound (11) follows from (87) by choosing,

$$\epsilon^* = \frac{\epsilon}{D^* \sum_{\lambda=0}^{N-1} (C^*)^{\lambda}}.$$

$\square$

The proof of proposition 4.5 can be readily extended to prove the following proposition about a general $r$-scale dynamical system of the form,

$$\begin{aligned}
\phi_n^1 &= \tau_1 \mathbf{f}^1(\phi_{n-1}^1, \phi_{n-1}^2, \ldots, \phi_{n-1}^r \mathbf{u}_n), \\
\phi_n^2 &= \tau_2 \mathbf{f}^2(\phi_{n-1}^1, \phi_{n-1}^2, \ldots, \phi_{n-1}^r \mathbf{u}_n), \\
&\cdots\cdots\cdots\cdots\cdots\cdots\cdots\cdots\cdots\cdots\cdots\cdots\cdots\cdots\cdots\cdots \\
&\cdots\cdots\cdots\cdots\cdots\cdots\cdots\cdots\cdots\cdots\cdots\cdots\cdots\cdots\cdots\cdots \\
\phi_n^r &= \tau_r \mathbf{f}^r(\phi_{n-1}^1, \phi_{n-1}^2, \ldots, \phi_{n-1}^r \mathbf{u}_n), \\
\mathbf{o}_n &= \mathbf{o}(\phi_n^1, \phi_n^2, \ldots, \phi_n^r).
\end{aligned} \tag{95}$$

Here, $\tau_1 \leq \tau_2 \ldots \ldots \leq \tau_r \leq 1$, with $r > 1$, are the $r$-time scales of the dynamical system (95). We assume that the underlying maps $\mathbf{f}^{1,2,\ldots,r}$ are Lipschitz continuous. We can prove the following proposition,

**Proposition E.4.** *For all $1 \leq n \leq N$, let $\phi_n^{1,2,\ldots,r}, \mathbf{o}_n$ be given by the $r$-scale dynamical system (95) with input signal $\mathbf{u}_n$. Under the assumption that there exists a $R > 0$ such that $\max\{\|\phi_n^1\|, |\phi_n^2\|, \ldots, |\phi_n^r\|, \|\mathbf{u}_n\|\} < R$, for all $1 \leq n \leq N$, then for any given $\epsilon > 0$, there exists a LEM of the form (3), with hidden states $\mathbf{y}_n, \mathbf{z}_n$ and output state $\omega_n$ with $\omega_n = \mathcal{W}\mathbf{y}_n$ such that the following holds,*

$$\|\mathbf{o}_n - \omega_n\| \leq \epsilon, \quad \forall 1 \leq n \leq N. \tag{96}$$

*Moreover, the weights, biases and size (number of neurons) of the underlying LEM (3) are* independent *of the time-scales* $\tau_{1,2,\ldots,r}$.

# F  LEMS EMULATE HETEROGENEOUS MULTISCALE METHODS FOR ODES

Following Kuehn (2015), we consider the following prototypical example of a fast-slow system of ordinary differential equations,

$$\begin{aligned} \phi'(t) &= \frac{1}{\tau}\left(f(\psi) - \phi\right), \\ \psi'(t) &= g(\phi, \psi). \end{aligned} \tag{97}$$

Here $\phi, \psi \in \mathbb{R}^m$ are the fast and slow variables respectively and $0 < \tau << 1$ is a small parameter. Note that we have rescaled time and are interested in the dynamics of the slow variable $\psi(t)$ in the time interval $[0, T]$.

A naive time-stepping numerical scheme for (97) requires a time step size $\delta t \sim \mathcal{O}(\tau)$. Thus, the computation will entail time updates $N \sim \mathcal{O}(1/\tau)$. Hence, one needs a multiscale ODE solver to approximate the solutions of the system (97). One such popular ODE solver can be derived by using the Heterogenous multiscale method (HMM); see Kuehn (2015) and references therein. This in turns, requires using two time stepping schemes, a *macro* solver for the slow variable, with a time step $\Delta t$ of the form,

$$\psi_n = \psi_{n-1} + \tilde{\Delta}tg(\phi_n, \psi_{n-1}). \tag{98}$$

Here, the time step $\tilde{\Delta}t < 1$ is independent of the small parameter $\tau$.

Moreover, the fast variable is updated using a *micro* solver of the form,

$$\begin{aligned} \phi_{n-1}^{(k)} &= \phi_{n-1}^{(k-1)} - \delta t(f(\psi_{n-1}) - \phi_{n-1}^{(k-1)}), \quad 1 \leq k \leq K. \\ \phi_n &= \phi_{n-1}^K, \\ \phi_{n-1}^{(0)} &= \phi_{n-1}. \end{aligned} \tag{99}$$

Note that the micro time step size $\delta t$ and the number of micro time steps $K$ are assumed to independent of the small parameter $\tau$.

It is shown in Kuehn (2015) (Chapter 10.8) that for any given small tolerance $\epsilon > 0$, one can choose a macro time step $\tilde{\Delta}t$, a micro time step $\delta t$, the number $K$ of micro time steps, the number $N$ of macro time steps, independent of $\tau$, such that the discrete states $\psi_n$ approximate the slow-variable $\psi(t_n)$ (with $t_n = n\tilde{\Delta}t$) of the fast-slow system (97) to the desired accuracy of $\epsilon$.

Our aim is to show that we can construct a LEM of the form (3) such that the states $\phi_n, \psi_n$, defined in (98), (99) can be approximated to arbitrary accuracy. By combining this with the accuracy of HMM, we will prove that LEMs can approximate the solutions of the fast-slow system (97) to desired accuracy, independent of the small parameter $\tau$ in (97).

**Proposition F.1.** *Let $\phi_n, \psi_n \in \mathbb{R}^m$, for $1 \leq n \leq N$, be the states defined by the HMM dynamical system (98), (99). For any given $\epsilon > 0$, there exists a LEM of the form (3) with hidden states $[\mathbf{z}_n, \mathbf{y}_n]$, where $\mathbf{z}_n, \mathbf{y}_n \in \mathbb{R}^{dm}$ and output states $\omega_n^h, \omega_n^c$ such that the following holds,*

$$\max\left\{\|\phi_n - \omega_n^h\|, \|\psi_n - \omega_n^c\|\right\} \leq \epsilon, \quad \forall 1 \leq n \leq N. \tag{100}$$

*Proof.* We start by using iteration on the micro solver (99) from $k = 1$ to $k = K$ to derive the following,

$$\phi_n = \overline{\delta t}\phi_{n-1} + (1 - \overline{\delta t})f(\psi_{n-1}),$$
$$\overline{\delta t} = (1 - \delta t)^K. \tag{101}$$

As $\delta t < 1$, we have that $\overline{\delta t} < 1$.

By the universal approximation theorem for tanh neural networks, for any given tolerance $\epsilon^*$, there exist weight matrices $W_1^f \in \mathbb{R}^{d_1 \times m}, W_2^f \in \mathbb{R}^{m \times d_1}$ and bias vector $b_1^f \in \mathbb{R}^{d_1}$ such that the tanh neural network defined by,

$$\mathcal{N}_f(c) = W_2^f \sigma \left( W_1^f c + b_1^f \right), \tag{102}$$

approximates the underlying function $\mathbf{f}$ in the following manner,

$$\max_{\|c\| < R^*} \|\mathbf{f}(c) - \mathcal{N}_f(c)\| \le \epsilon^*. \tag{103}$$

Next, we define the following map,

$$\mathbf{G}(h, c) = \mathbf{g}(h, c) + c, \tag{104}$$

By the universal approximation theorem, given $\epsilon^*$, there exist weight matrices $W_1^g, W_2^g \in \mathbb{R}^{d_2 \times m}, W_3^g \in \mathbb{R}^{m \times d_2}$ and bias vector $b_1^g \in \mathbb{R}^{d_2}$ such that the tanh neural network defined by,

$$\mathcal{N}_g(h, c) = W_3^g \sigma \left( W_1^g h + W_2^g c + b_1^g \right), \tag{105}$$

approximates the function $\mathbf{G}$ (104) in the following manner,

$$\max_{\max(\|h\|, \|c\|) < R^*} \|\mathbf{G}(h, c) - \mathcal{N}_g(h, c)\| \le \epsilon^*. \tag{106}$$

Next, as in the proof of propositions 4.4 4.5, one can readily approximate the identity function $\hat{f}(h, c) = h$ with a tanh neural network of the form,

$$\bar{\mathcal{N}}_f(h) = \bar{W}_2 \sigma \left( \bar{W}_1 h \right), \tag{107}$$

such that

$$\max_{\|h\|, \|c\| < R^*} \|\hat{f}(h, c) - \mathcal{N}_f(h)\| \le \epsilon^*, \tag{108}$$

and with the same weights and biases, one can approximate the identity function $\hat{g}(h, c) = c$ with the Tanh neural network,

$$\bar{\mathcal{N}}_g(c) = \bar{W}_2 \sigma \left( \bar{W}_1 c \right), \tag{109}$$

such that

$$\max_{\|h\|, \|c\| < R^*} \|\hat{g}(h, c) - \mathcal{N}_g(c)\| \le \epsilon^*. \tag{110}$$

Then, we define the following dynamical system,

$$\begin{aligned}
\hat{\mathbf{z}}_n &= \overline{\delta t}\hat{\mathbf{z}}_n + (1 - \overline{\delta t})W_2^f \sigma \left( W_1^f \hat{\mathbf{y}}_{n-1} + b_1^f \right), \\
\tilde{\mathbf{z}}_n &= \bar{W}_2 \sigma \left( \bar{W}_1 \hat{\mathbf{y}}_{n-1} \right), \\
\hat{\mathbf{y}}_n &= (1 - \tilde{\Delta}t)\hat{\mathbf{y}}_{n-1} + \tilde{\Delta}t W_3^g \sigma \left( W_1^g \hat{\mathbf{z}}_n + W_2^g \tilde{\mathbf{z}}_n + b_1^g \right), \\
\tilde{\mathbf{y}}_n &= \bar{W}_2 \sigma \left( \bar{W}_1 \hat{\mathbf{z}}_n \right),
\end{aligned} \tag{111}$$

with hidden states $\hat{\mathbf{z}}_n, \tilde{\mathbf{z}}_n, \hat{\mathbf{y}}_n, \tilde{\mathbf{y}}_n \in \mathbb{R}^m$ and with initial states $\hat{\mathbf{z}}_0 = \tilde{\mathbf{z}}_0 = \hat{\mathbf{y}}_0 = \tilde{\mathbf{y}}_0 = 0$.

Completely analogously as in the derivation of (87), we can derive the following bound,

$$\|\phi_n - \hat{\mathbf{z}}_n\| + \|\psi_n - \hat{\mathbf{y}}_n\| \le C^* \epsilon^*, \tag{112}$$

with constant $C^* = C^* (n, \text{Lip}(f), \text{Lip}(g))$.

Defining the dynamical system,

$$
\begin{aligned}
\mathbf{z}_n^* &= \overline{\delta t}\mathbf{z}_n^* + (1 - \overline{\delta t})\sigma\left(W_1^f W_3^g \hat{\mathbf{y}}_{n-1} + b_1^f\right) \\
\bar{\mathbf{z}}_n &= \sigma\left(\bar{W}_1 W_3^g \mathbf{y}_{n-1}^*\right) \\
\mathbf{y}_n^* &= (1 - \tilde{\Delta}t)\mathbf{y}_{n-1}^* + \tilde{\Delta}t\sigma\left(W_1^g W_3^f \mathbf{z}_n^* + W_2^g \bar{W}_2 \tilde{\mathbf{z}}_n + b_1^g\right) \\
\bar{\mathbf{y}}_n &= \sigma\left(\bar{W}_1 W_2^f \mathbf{z}_n^*\right).
\end{aligned}
\tag{113}
$$

By multiplying suitable matrices to (113), we obtain that,

$$
\hat{\mathbf{z}}_n = W_2^f \mathbf{z}_n^*, \quad \tilde{\mathbf{z}}_n = \bar{W}_2 \bar{\mathbf{z}}_n, \quad \hat{\mathbf{y}}_n = W_3^g \mathbf{y}_n^*, \quad \tilde{\mathbf{y}}_n = \bar{W}_2 \bar{\mathbf{y}}_n.
\tag{114}
$$

In addition to $b_\infty$ defined in (58), for $\overline{\delta t} \in (0, 1]$, we introduce $b_\delta \in \mathbb{R}$ defined by

$$
\hat{\sigma}(b_\delta) = \overline{\delta t}.
\tag{115}
$$

Similarly for $\tilde{\Delta}t \in (0, 1]$, we introduce $b_\Delta \in \mathbb{R}$ defined by

$$
\hat{\sigma}(b_\Delta) = \tilde{\Delta}t.
\tag{116}
$$

The existence of unique $b_\delta$ and $b_\Delta$ follows from the fact that the sigmoid function $\hat{\sigma}$ is monotone.

Next, we define the two vectors $\mathbf{b}_\infty, \mathbf{b}_\delta, \mathbf{b}_\Delta \in \mathbb{R}^{2m}$ as

$$
\begin{aligned}
\mathbf{b}_\delta^i &= b_\delta, \quad \forall\, 1 \le i \le m, \\
\mathbf{b}_\delta^i &= b_\infty, \quad \forall\, m+1 \le i \le 2m, \\
\mathbf{b}_\Delta^i &= b_\Delta, \quad \forall\, 1 \le i \le m, \\
\mathbf{b}_\Delta^i &= b_\infty, \quad \forall\, m+1 \le i \le 2m.
\end{aligned}
\tag{117}
$$

We define the LEM of form (3), which will approximate the HMM (98),(99). To this end, we define the hidden states $\mathbf{z}_n, \mathbf{y}_n \in \mathbb{R}^{2m}$ such that $\mathbf{z}_n = [\mathbf{z}_n^*, \bar{\mathbf{z}}_n^*]$ and $\mathbf{y}_n = [\mathbf{y}_n^*, \bar{\mathbf{y}}_n^*]$. The parameters for the corresponding LEM of form (3) given by,

$$
\begin{aligned}
&\Delta t = 1, d_y = 2m \\
&\mathbf{W}_1 = \mathbf{W}_2 = \mathbf{V}_1 = \mathbf{V}_2 \equiv 0, \\
&\mathbf{b}_1 = \mathbf{b}_\delta, \quad \mathbf{b}_2 = \mathbf{b}_\Delta, \\
&\mathbf{W}_z = \begin{bmatrix} W_1^f W_3^g & 0 \\ \bar{W}_1 W_3^g & 0 \end{bmatrix}, \quad \mathbf{V}_z = 0, \quad \mathbf{b}_z = [b_1^f, 0], \\
&\mathbf{W}_y = \begin{bmatrix} W_1^g W_3^f & W_2^g \bar{W}_2 \\ \bar{W}_1 W_2^f & 0 \end{bmatrix}, \quad \mathbf{V}_z = 0, \quad \mathbf{b}_z = [b_1^g, 0].
\end{aligned}
\tag{118}
$$

The output states are defined by,

$$
\omega_n^h = W_2^f \mathbf{z}_n^*, \quad \omega_n^h = W_3^g \mathbf{y}_n^*
\tag{119}
$$

It is straightforward to observe that $\omega_n^h = \hat{\mathbf{z}}_n$, $\omega_n^c = \hat{\mathbf{y}}_n$. Hence, the desired bound (100) follows from (112) by choosing,

$$
\epsilon^* = \frac{\epsilon}{C^*}.
$$

$\square$

