# OpenReview forum: "Long Expressive Memory for Sequence Modeling"
_ICLR.cc/2022/Conference — ICLR 2022 Spotlight_

### Official Review · Reviewer_Cggm · 2021-11-01

**Correctness:** 4
**Technical Novelty And Significance:** 3
**Empirical Novelty And Significance:** 3
**Recommendation:** 6
**Confidence:** 3

**Main Review:**

Strengths:

1. They introduce an architectural gadget for sequence modeling that appears to a) add multi-scale sequence modeling explicitly in the model, b) (somewhat) stabilize gradients in training, and c) be very expressive.
2. They justified their claims with experiments that demonstrate the benefits of the architecture on some small datasets.
3. They proved some expressivity theorems and bounded the size of the resulting gradients.


Comments:

1. What is meant by "LEM is gradient-based" and "gradient-based architecture"? I would refer to the model in the way I did in the summary of the paper -- you're giving a circuit for the model architecture which is defined by discrete time updates of a system of ODEs.
2. What is meant by an "algorithm for your model" (in the line "In order to realize a concrete algorithm for our model")? (see above, circuit is probably a better word here).
3. I found the notation for X_n unnecessarily complicated. Just say X_n = [x_n, y_n]. Likewise just define E_n as (1/2) \|y_n - y_n*\|_2^2.
4. Originally, before reading Remark D.1, I wrote the following: "I don't follow the conclusion after Prop 4.2. You just proved a bound on the gradient that depends on the l-infinity norm of (y*, z*), which in Prop 4.1. you showed to be bounded by Theta(sqrt(n)) (where n is the number of iterations). So it seems like the gradient is bounded to grow in l-infinity norm with at most sqrt(n), but as n -> infinity this certainly blows up, just at a sublinear rate. Am I missing something? I see later you assume that the ground truth (y*, z*) norms are constant; why is this a reasonable assumption? I also see that later on in the supplement, you say that t_n <= 1 -- I don't follow why this is. (Also this should be stated in the main text). Aren't you just choosing N bounded in that case, in which case yes, by default anything that grows with N will be bounded?" Two points after this: 1) You should make these points clearer in the main text; 2) I see it actually grows as a polynomial of T -- why are you saying you can simply rescale T? It seems to me that if the weights/gradients grow with polynomially with T, you have not actually achieved bounded gradients -- this is my main issue with the paper so far, so an explanation would be appreciated. Is it just that the growth is not exponential with T? If that's the point, it should be emphasized.

5. In Eq. 7, should it be Theta instead of big O? Also why include the square term when it is dominated by the 3/2 term?

**Summary Of The Paper:**

This paper introduces a model class of neural networks whose architecture is defined by a circuit that computes a discretized step of a pair of multi-scale ODEs. The key novelty is on the multi-scale aspect of this system. In terms of representation capability, this class can represent LSTMs (and vice-versa) and can also approximate any dynamical system from a very broad class of dynamical systems (as well as multi-scale dynamical systems). The authors also prove upper bounds in l_infinity on both the hidden state values and the gradients, and argue this addresses the exploding gradient problem. They also address the vanishing gradient problem by arguing that the magnitude of the partial gradient corresponding to the contribution that depends on step k of the circuit does not depend on k.

They then present a variety of experiments using this new architectural component for modeling several sequence problems. They include a small study on the multi-scale nature of the datasets they study as well.

**Summary Of The Review:**

Overall, I recommend accept conditional on satisfactory explanations to some of my questions -- it's an interesting, relatively principled derivation of an architectural gadget for long range sequence modeling, building on the recurrent network modeling approach rather than the popular attention based approach. I like the idea of deriving the architectural updates from ODEs which have the desired properties you want. I thought the theorems were not that strong, but fine in terms of giving a little motivation for the architecture. Overall the experiments were good, though they tended to only be on rather small datasets.


UPDATE:

Thanks for the clarifications, I updated my correctness score and still vote to accept.

---

> ### Author Response · Authors · 2021-11-19
> **Reply to Reviewer Cggm - Part 1**
>
> We start by thanking the reviewer for your appreciation of the merits of our paper and your welcome suggestions to improve it. Below, we address the very valid concerns raised by the reviewer and thank the reviewer in advance for their patience in reading our detailed reply.
> * The term "gradient-based" was motivated by its use in the original LSTM paper of Hochreiter and Schmidhuber. We retain its use in a few places while also incorporating your excellent suggestion of interpreting the architecture in terms of a *circuit* defined by a discretized systems of ODEs.
> * The phrase "algorithm for the model" has been modified in the revised version.
> * We followed your suggestion and changed the notation for ${\bf X}_n$ and the loss function $\mathcal{E}_n$ in the revised version.
> * Your concerns about the flow of arguments for our theoretical bounds are very valid and we apologize for the possible confusion stemming from our write-up, particularly with respect to unnecessary notation and assumptions. We have now rewritten the sections on the bounds (and their proofs) and proceed to address the specific concerns that you raise below
>    - Given that we are modeling sequences, the relevant input is the sequence length $n$. ${\Delta t}$ is a hyperparameter that we can choose and in practice is determined by a simple hyperparameter tuning procedure detailed in **SM** Section A.
>    - We start with the bound (4) on the hidden states $\bf y_n$, $\bf z_n$. Under the assumption that $\Delta t < 2$, we have now obtained a precise bound (4) on the $\ell_{\infty}$-norm of the hidden state vectors in Proposition 4.1 of the main text. This bound scales as ${\mathcal O}(\sqrt{t_n})$ with $t_n = n \Delta t$. Simply setting $\Delta t \sim \frac{1}{n}$ implies that the hidden states grow as ${\mathcal O}(1)$ and are bounded independently of the sequence length $n$. This choice of $\Delta t$ suffices for theoretical purposes but might not yield the best results in practice. As examined extensively in Section $B.1$ of **SM**, a choice of $\Delta t \sim \frac{1}{\sqrt{n}}$ seems to be optimal for problems with long-range dependencies. Consequently, the hidden states can grow as ${\mathcal O}(n^{\frac{1}{4}})$. Such slow growth seems to be consistent with previous empirical results in the literature; see for instance Figure 4 (iii) of reference [R1], where slow growth of hidden states, with respect to sequence length, has been argued as superior to bounded hidden states.
>    - We use the definition of the exploding and vanishing gradient problem as given in reference [R2] (and references therein), which is widely accepted as the standard definition in this field. As [R2] defines it, the partial gradient $\partial_\theta E^{(k)}_n$ (defined in (5)) measures the contribution to the hidden state gradient at step $n$ arising from the hidden state at step $k \leq n$ of the model. This partial gradient can behave as $\partial_\theta E^{(k)}_n \sim \gamma ^{n-k}$, for some $\gamma > 0$. If $\gamma > 1$, then the gradient can grow exponentially with $n$ and the model is defined to incur the exploding gradient problem. On the other hand, if $\gamma < 1$, then the gradient can decay exponentially with $n$ and the model is defined to incur the vanishing gradient problem. We clearly state these considerations in the main text now.
>    - In proposition 4.2 of the main text, we obtain an explicit bound on the total gradient $\partial_\theta E_n = \sum\limits_{k=1}^n \partial_\theta E^{(k)}_n$ (with proof in Section E.2). This estimate (6) provides a precise upper bound on the total gradient in terms of powers of $t_n = n \Delta t$. Again choosing $\Delta t \sim \frac{1}{n}$ ensures that the total gradient is bounded above, uniformly in sequence length. As in the case of hidden states, setting $\Delta t \sim \frac{1}{\sqrt{n}}$ leads to a bound that scales as $n^{\frac{3}{2}}$ (see the bound (41) in **SM** Remark E.1 for the general case). In either case, the partial gradient $\partial_\theta E^{(k)}_n$ scales as $\partial_\theta E^{(k)}_n \sim {\mathcal O}(1)$ (for $\Delta t \sim \frac{1}{n}$) or $\partial_\theta E^{(k)}_n \sim {\mathcal O}(n^{\frac{1}{4}})$ (for $\Delta t \sim \frac{1}{\sqrt{n}}$) (see bound (40) in **SM**). Thus, for these choices of $\Delta t$, the partial gradient can at most grow polynomially (even sub-linearly) in the sequence length $n$, well below the exponential growth that characterizes the exploding gradient problem. Hence, we claim that the exploding gradient problem is mitigated for LEM. We hope to have clarified the valid issues raised by you in this context and would like to thank you for motivating us to sharpen our bounds and the resulting presentation of them.
> * We have now truncated the expansion to leading order in (7) as suggested by you.

---

> > ### Author Response · Authors · 2021-11-19
> > **Reply to Reviewer Cggm - Part 2**
> >
> > ## References
> > [R1] Martin Arjovsky, Amar Shah, and Yoshua Bengio. Unitary evolution recurrent neural networks. In
> > International Conference on Machine Learning, pp. 1120–1128, 2016.
> >
> > [R2] Razvan Pascanu, Tomas Mikolov, and Yoshua Bengio. On the difficulty of training recurrent neural
> > networks. In Proceedings of the 30th International Conference on Machine Learning, volume 28
> > of ICML 13, pp. III–1310–III–1318. JMLR.org, 2013.

---

### Official Review · Reviewer_AvfW · 2021-11-03

**Correctness:** 4
**Technical Novelty And Significance:** 3
**Empirical Novelty And Significance:** 3
**Recommendation:** 6
**Confidence:** 3

**Main Review:**

The authors give an accessible overview of the variety of RNN modifications already proposed to help capture long-range dependencies. It would be useful to clarify somewhat how their approach (LEM) differs from the variety of ODE-based methods listed in Section 3. I would also note that there exist approaches to RNN analysis and modification entirely outside the 'vanishing gradient" framework, e.g. those based on statistical definitions of long-range dependence.

The LEM model is itself clearly motivated and introduced. It is somewhat unclear, however, what happens to the interpretation of $y$ and $z$ (originally slow and fast variables, respectively) when the timescales are parameterized and learned. Instead, as discussed in SM$\S$C, the interpretation seems much closer to that of the cell and hidden state in an LSTM (also: slight notational typo in the final line of Eq (15)). The main difference to LSTM appears to be a more sophisticated recursion on the hidden state and the introduction of the timescale hyperparameter $\Delta t$.

The experiments are extensive and show that LEM consistently achieves better performance on a range of tasks than several recent and state-of-the-art competitor RNN architectures. On the other hand, it seems somewhat concerning that both model performance in practice and some of the analytical results appear sensitive to the choice of $\Delta t$. Choosing a small $\Delta t$ practically guarantees that the cell and hidden states will persist over long timescales; rather than being learned, this behavior is directly selected as a hyperparameter. How is this different from adding a hyperparameter to LSTM to control the range of the input and forget gates?

Finally, the notion of multiple timescales is claimed as central to the motivation for the method proposed, but despite the extensive experimentation relatively little is done to understand whether and how such behavior arises in the trained LEM models - in fact only a few sentences on this topic appear outside the supplement. There are some questions here that seem relevant to understanding this method in greater detail:
- What is the interpretation of results in SM$\S$A? The observation of a broad range of timescales does not necessarily imply that they are important for model performance. There might be some relatively straightforward ablation experiments here (e.g. clipping or removing this variation in timescales) that could address this question more thoroughly.
- The power law behavior of timescale frequencies is interesting, but doesn't its slope suggest that the trained models in fact place very little emphasis on representing changes over long timescales? Note that this is exactly the opposite behavior of long memory stochastic processes, which have power-law spectral density functions that *increase* in power at low frequencies.
- To the extent that the relevant timescale(s) vary across the variety of tasks introduced, is this observed in the learned timescales of the LEM models?

Other comments:
- Prop. 4.2: Why would there be a "ground truth" cell state $z$? It makes sense for $y$ as that's the prediction target, but a ground truth $z$ seems to suggest some assumption on how the data is generated.
- I don't understand the "ablation study" reported in Sec. G of the Supplement. What exactly is being ablated? The figure seems to show test accuracy as a function of $\Delta t$.
- The statement in Section 6 that natural language and speech data do not necessarily contain long-range dependencies is contrary to quite a large body of empirical and statistical analysis of such data.

**Summary Of The Paper:**

The authors propose a new RNN architecture, long expressive memory (LEM), motivated by a system of ODEs with multiple time constants. They prove that it can avoid the vanishing gradient problem while retaining the flexibility to approximate a broad class of dynamical systems. They report comparable or improved prediction performance of LEM-based sequence models across a very wide variety of tasks, as compared to several recent alternatives.

**Summary Of The Review:**

The proposed method is clearly presented and thoroughly demonstrated in terms of prediction performance across many tasks. However, there is surprisingly little investigation as to whether it learns representations at multiple timescales or how important these are to its success, and there is some uncertainty as to the sensitivity of results with respect to the hyperparameter $\Delta t$. My score reflects these weaknesses but could be improved if they are sufficiently addressed or rebutted.

UPDATE: Following the authors' responses to this review and others, I have improved my evaluation of the technical significance and my overall recommendation.

---

> ### Author Response · Authors · 2021-11-19
> **Reply to Reviewer AvfW - Part 1**
>
> We start by thanking the reviewer for your appreciation of the merits of our paper and your welcome suggestions to improve it. Below, we address the very valid concerns raised by the reviewer and thank the reviewer in advance for their patience in reading our detailed reply.
> * In **SM** Section D, we have now added further connections between LEM and LSTM. In particular and as pointed out by you, we highlight the fact that LEM has a sophisticated recursion defining the update of each cell state. Many thanks for pointing out the minor typo in (15) - it is corrected now. We have also written about the distinction between LEM and other ODE based RNNs in Section 3.
> * On the role of the hyperparameter $\Delta t$:
>    - As you point out, our theoretical bounds on the hidden states (4) and hidden state gradients (6) do require a small enough value of $\Delta t$, for instance $\Delta t \sim \frac{1}{n}$ (with $n$ being the sequence length) suffices to obtain uniform (in $n$) bounds on the hidden states and their gradients. However, we would like to point that these upper bounds are worst-case guarantees and may be quite pessimistic for a given problem at hand. Hence, one should tune the hyperparameter $\Delta t$ by a fairly standard tuning procedure: details in **SM** Section A, where the resulting values for $\Delta t$ for all our experiments are shown in **SM** Table 8. We observe from this table that for many tasks, no tuning is necessary and a default value of $\Delta t =1$ suffices. For problems with very long sequence lengths, a more careful analysis of the empirical data, now presented in **SM** Section $B.1$ reveals that a $\Delta t$ in the range of $\frac{1}{\sqrt{n}} \leq \Delta t \leq 1$ suffices in practice for obtaining optimal empirical performance. Given this, even for the EigenWorms experiment with very long sequence length of $n=17984$, a $\Delta t \sim 0.01$ was enough to provide state of the art accuracy. Thus, in practice, these values of $\Delta t$ suggest that the hidden states are not static during the training period and the model learns fairly quickly. Consequently, we were able to obtain very high accuracy for the Eigenworms experiments with only $50$ epochs of training. Across the board (see Section A and Figure 3 of **SM** and Figure 1 of main text), we observe that relatively few training iterations are necessary for the model to learn, independent of the underlying input sequence length.
>    - Your suggestion about adding a hyperparameter $\Delta t$ to the input and forget gates of an LSTM is very interesting and we decided to investigate if this can improve LSTM baselines significantly. To this end, we considered two alternatives **1)** scale (multiply) the input ${\bf i}_n$ and forget ${\bf f}_n$ gates in the LSTM (15) with the hyperparameter $\Delta t$ or **2)** Based on relation between LSTM and LEM (see **SM** Section D), we multiply $\Delta t$ to the input gate ${\bf i}_n$ of the LSTM (15) and $1-\Delta t$ to the forget gate ${\bf f}_n$ of the LSTM (15). We tuned with the hyperparameter $\Delta t$, in the range $0 < \Delta t \leq 1$ with exactly the same tuning procedure as for LEM and tested this LSTM variant on the EigenWorms experiment. The resulting test accuracies (see Table 11 in **SM**) were $53.3\%$ for variant 1 and $56.9\%$ for variant 2, which definitely improved the LSTM baseline $38.5\%$ but were well below the accuracy of LEM ($92.3\%$) and other competing models (Table 2). Thus, we demonstrate that just rescaling LSTM with a small hyperparameter is not sufficient to significantly improve LSTM performance on tasks with very-long dependencies. These findings are not surprising as LEM is not just a rescaled LSTM, but as you correctly identify, it has a very different recursive update, as compared to LSTM. Moreover, even when $\Delta t=1$ in LEM as in the FitzHugh-Nagumo task, Google12 task and character and word level PTB, it significantly outperforms LSTM as shown in Tables 4,5,6 and 7. We hope that these experiments clearly bring out the strengths of LEM in comparison to LSTM.

---

> > ### Author Response · Authors · 2021-11-19
> > **Reply to Reviewer AvfW - Part 2**
> >
> > * On the relevance of multiple scales in LEM: Your concerns regarding multiple scales are certainly valid and we apologize for our possibly unclear presentation of how we think the multiscale resolution of LEM impacts its performance. Based on your questions and suggestions, we have rewritten our arguments about the role of multiple scales in LEM and presented them in **SM**, Section $B.2$. Our detailed clarifications to your specific points are listed below,
> >    - LEM is indeed designed to express multiple scales and a clear  indication of the advantages in doing so are given in Proposition 4.4 of the main text, where we show that LEM can approximate a very general two-scale dynamical system (10) to arbitrary accuracy. This result can be readily extended to a dynamical system (92) with $r \geq 2$ time scales. We have added Proposition E.3 (in **SM** Section E.5) to show that this $r$-scale system can be approximated by LEM to arbitrary accuracy. Moreover, the weights and biases of the constructed LEMs in Propositions 4.4 and E.3 are *independent* of the underlying $r$-scales $\tau_1,\ldots,\tau_r$. Thus in principle, the multiscale structure of LEM enables it to approximate dynamical systems with multiple scales.
> >    - Given these theoretical results, it is natural to ask if in practice, LEM can approximate such systems and does so by deploying its multiscale features. To this end, we considered the FitzHugh-Nagumo system (12), which is a well-known prototype of two-scale (fast-slow) dynamical system. In our experiments, we fix the two time scales as $\tau_1 = 0.02$ and $\tau_2 =1$. From Table 4, we have already observed that LEM provides significantly lower test errors on this task than competing models. To see if the multiscale features of LEM were expressed, we plot the distribution of learned scales ${\bf \Delta t}_n$ and ${\bf \overline{\Delta t}}_n$ over the sequence length $n$ and over all hidden units and present the resulting empirical histogram in **SM** Figure 4. As shown in this plot, the learned scales range in amplitude over two orders of magnitude, but with a clear concentration near $\tau_1 = 0.02$ and $\tau_2=1$. Thus, the distribution of learned scales in LEM for this problem reflects the underlying time scales $0.02$ and $1$ very well. Hence, we conclude that for this multiscale dynamical system, LEM does what it is designed to do, i.e., it learns the underlying multiple time scales and achieves superior performance. Given these results and your comments in this direction, we proceeded to check if the quality of results would be affected by disabling (ablating) the multiscale feature of LEM. To this end, we simply set ${\bf \Delta t}_n= {\bf \overline{\Delta t}}_n \equiv \Delta t{\bf 1}$, for all $n$ ($\bf 1$ being the vector with all entries set to $1$), with $\Delta t$ being the tunable hyperparameter. We tuned it in the same manner as for LEM and observe a significant drop in accuracy, namely the test error increased from $0.002$ in LEM to $0.011$ in the single scale version of LEM (which is comparable to the LSTM error of $0.012$), see **SM** Table 9. Thus, we established that the multiscale resolution of LEM is essential for its superior performance on this task.
> >    - As you rightly ask: Does this multiscale resolution of LEM hold for a variety of tasks ? To investigate this, we consider the Google12 keyword spotting experiment and plot the distribution (empirical histogram) of the learned scales ${\bf \Delta t}_n$ and ${\bf \overline{\Delta t}}_n$ in **SM** Figure 5. Again, a range of scales, amounting to $2-3$ orders of magnitude in scale amplitude are expressed in the trained LEM for this task. But are these range of scales necessary ? Following your excellent suggestion, we performed a study where we *clipped* the range of the learned scales as $2^{-i} \leq {\bf \Delta t}_n, {\bf \overline{\Delta t}}_n \leq 1$, with $i=0,1,2,\ldots,7$ and plot the results in **SM** Figure 6. We observe from this figure that the accuracy is very poor (below $65\%$) for $i \leq 3$ but jumps to around $90\%$ when the learned scales are allowed to range more than an order of magnitude. The accuracy increases monotonically till the full range of scales (more than two orders of magnitude) is allowed to be expressed. Thus, this experiment clearly indicates that not only does LEM learn multiple scales, they seem to be essential for its observed superior performance. This is reinforced by the empirical histogram for the distribution of ${\bf \Delta t}_n$ and ${\bf \overline{\Delta t}}_n$ (**SM** Figure 5), where the occurrence of small scales decays as a power law of the scale amplitude, rather than exponentially, indicating that the small scales do contribute to the overall dynamics. The same power law behavior is also observed in the sMNIST experiment (see **SM** Figure 7), indicating the expression of multiple scales and their importance holds across many diverse tasks.

---

> > > ### Author Response · Authors · 2021-11-19
> > > **Reply to Reviewer AvfW - Part 3**
> > >
> > > * **Continuation of** "On the relevance of multiple scales in LEM":
> > >    - We apologize for the possible confusion caused by the Y-axis labels on our histograms. In **SM** Figures 4, 6 and 7, we are simply plotting the histogram, representing the distribution of learned scales ${\bf \Delta t}_n$ and ${\bf \overline{\Delta t}}_n$ and not a power spectrum for a stochastic process. You also point out an interesting connection with *long memory processes*. In our understanding, the time-autocorrelation function of such long memory processes decays as power law of the time increment. The question of investigating if LEM can learn such long memory processes accurately is very interesting. Given the time and page limit constraints, we defer this avenue for future investigation and thank you for pointing it out to us.
> > > * There was a typo in Propositon 4.2. As you correctly pointed out, the bound only depends on the ground truth $\overline{\bf Y}$ and not on some ground-truth state $\overline{\bf Z}$. This is now corrected.
> > > * Our apologies for the incorrect use of the term "ablation". We wanted to report the sensitivity of the results to the choice of hyperparameter $\Delta t$ and have now removed ablation in this context.
> > > * Our claim on long-term dependencies in the context of language modeling could be misleading. We have removed it from the revised version of the article.

---

> > > > ### Comment · Reviewer_AvfW · 2021-11-28
> > > > **Follow-up on author responses**
> > > >
> > > > Thanks to the authors for their thoughtful and detailed response to my original review. I have re-read the updated parts of the paper and supplement, along with the discussion posted by the authors in response to each review.
> > > >
> > > > The additional experiments in **SM**$\S$B, along with their discussion above, directly address what was my main criticism of this manuscript and provide evidence that the learned timescales in the LEM model both (a) can identify known timescales in the data in toy examples and (b) outperform in a predictive sense baselines in which their diversity is restricted or eliminated. There are many potential uses for a nonlinear time series model that explicitly represents and learns a mixture of timescales that optimize predictive accuracy. The authors' additions have clarified this contribution and improved the opportunity for the community to learn from and potentially extend this work.
> > > >
> > > > I am pleased to improve my score and recommend that this paper be accepted.

---

### Official Review · Reviewer_M6L8 · 2021-11-03

**Correctness:** 3
**Technical Novelty And Significance:** 4
**Empirical Novelty And Significance:** 3
**Recommendation:** 8
**Confidence:** 4

**Main Review:**

I found this paper well-written and relatively easy to read, considering the amount of theoretical and experimental results presented. I was able to follow the theoretical arguments well, though I did not check the proofs line-by-line. Intuitively, the proposed LEM is a type of LSTM with no output gates and two cell states per unit instead of one: $z_n$ and $y_n$. $\Delta t_n$ and $\overline{\Delta t_n}$ are the two corresponding input gates, and the forget gates are "coupled" to the input gates. Finally the cell inputs for $z$ are $y_{n-1}, u_n$ while those for $y$ are $z_n, u_n$. Thus the $y$ cell "sits above" the $z$ cell.

The design is new and interesting, and I can see the intuitive appeal (which is supported by the theoretical connections/results). The theoretical results related to vanishing/exploding gradients also seem intuitive: they are reliant on small enough values of $\Delta t$, which will bring both cells close to copy behavior (like in LSTMs) which propagates gradients well.

It is also a strength that the paper presents experimental results on a variety of benchmarks with different types of data, though it should be noted that all benchmarks are small-scale (which should be sufficient for the purposes of this paper).
The results are very good across the board, with LEM beating LSTM/GRU in particular across the board.

This brings me to my main objections to this paper. I am uncertain that the LSTM baselines are as strong as they should be for various tasks considered, and this might be misrepresenting the performance that can be achieved using LSTMs. I recommend that the authors consider the Chrono initialization [A] as a starting point, and obtain their own baseline results instead simply taking them from other papers that may or may not have been diligent in evaluating baselines. The results in that paper can not be directly compared to those here, but they are very suggestive: at least for $T$=750 on the adding problem, the LSTM was able to learn rapidly (in 500 to 1000 steps), so it would be surprising if it fails completely for longer lengths. This should also be relevant for the EigenWorms dataset etc. For pMNIST, the Chrono init paper reports a result of 96.3% on the val. set (compared to 92.9% reported in the present paper), although that is without tuning model size or hyperparameters.

In my opinion, the above observations make a strong case that the baseline results (at least for LSTM) might be too weak in this paper, potentially misleading the authors/readers into just how much of an improvement LEM provides in practice. Since this is a nice paper in other aspects, I hope that the authors will invest in addressing this issue so that I can increase my score (if the results remain interesting).

[A] Tallec, C., & Ollivier, Y. (2018). Can recurrent neural networks warp time?. arXiv preprint arXiv:1804.11188.

—
Update: The authors have addressed my concerns related to experimental results, so I am updating my score from 6 to 8. Of course more work is needed to generally prove the wider utility of the new architecture since all tasks here are small scale, but the initial results are interesting enough to warrant a wider examination by the community.

**Summary Of The Paper:**

This paper introduces Long Expressive Memory (LEM; Eq. 3) which is a new architecture for recurrent networks derived from discretization of a particular system of multi-scale ODEs (Eq. 2). It presents interesting theoretical results that characterize the properties of the proposed architecture (Sec. 4). The key results are that under certain conditions (e.g. fine discretization) the gradient propagation through time is well behaved, and that it is expressive enough to represent general dynamical systems, and multiscale dynamical systems in particular. Empirical results are presented on the adding problem, a synthetic two-scale dynamical system, sequential MNIST/CIFAR-10 classification, EigenWorms classification of very long time series, heart rate prediction, the Speech Commands dataset, and character-level modeling on Penn Treebank (Sec. 5). Across all tasks, the authors report best performance using the proposed architecture when compared to various methods in the literature.

**Summary Of The Review:**

The paper is well written and the presented architecture is intuitively appealing and supported by reasonable theoretical results. The experimental results are interesting but their significance is unclear since I am not convinced that the LSTM baselines are strong enough, based on prior work.

---

> ### Author Response · Authors · 2021-11-19
> **Reply to Reviewer M6L8**
>
> We start by thanking the reviewer for your appreciation of the merits of our paper and your welcome suggestions for improving it. Below, we address the very valid concerns raised by the reviewer and thank the reviewer in advance for their patience in reading our detailed reply.
> * We greatly appreciate your heuristic interpretation of the relation between LSTM and LEM and have now mentioned it in **SM** Section D.
> * We apologize for not being aware of the chrono initialization in Tallec and Ollivier 2018 and thank you for pointing this out. As you suggested, we have implemented the *chrono* initialization for LSTM on a variety of learning tasks and report these results in our paper. For the adding problem, the chrono-LSTM provides a very large improvement over baseline LSTM (see Figure 1) and now emerges as the main competitor to LEM on this problem, especially for an input sequence length of $n=10000$. Nevertheless, LEM does outperform chrono-LSTM in terms of a consistently lower test error as well as consistently lower number of training steps needed to attain this error. Moreover, we tested chrono-LSTM on every learning task except the PTB language modeling task where the authors of Tallec and Ollivier 2018 say that the chrono initialization does not significantly improve the results of the underlying baseline model. The chrono initialization significantly improved the LSTM results for most tasks with reasonably long sequence lengths as reported now in Tables 1,2 and 3. Nevertheless, LEM consistently outperformed chrono-LSTM on all these tasks. For instance on the Eigenworms dataset, the mean test accuracies were: Baseline-LSTM ($38.5\%$), Chrono-LSTM ($82.6\%$) and LEM ($92.3\%$). On the healthcare dataset, the test errors were: Baseline-LSTM ($9.93$), Chrono-LSTM ($3.31$), LEM ($0.85$). In this context, we would like to point out that for psMNIST, we have used $128$ hidden units for evaluating all the models as most of the baseline results in the literature use this number of units. As you point out, the chrono-LSTM has a reported accuracy of $96.3$\% with $512$ hidden units. Even with $128$ units, chrono initialization does improve the LSTM results from $92.9\%$ to $94.6\%$ (compared to an accuracy of $96.6\%$ for LEM). On the other hand, the chrono initialization did not lead to any noticeable improvement over the LSTM baseline for other tasks. For instance for FitzHugh-Nagumo multiscale dynamical systems prediction, the test errors (multiplied by 100) are Baseline-LSTM ($1.2$), chrono-LSTM ($2.0$) and LEM ($0.2$). Similarly for the Google12 dataset, the test accuracies are Baseline-LSTM ($94.9\%$), chrono-LSTM ($94.8\%$) and LEM ($95.7\%$). We chose not to add the results for chrono-LSTM when it did not improve the LSTM baseline.
> * Finally, given how much the chrono initialization improved the LSTM baselines and the relation between LSTM and LEM, we tried to adapt the chrono initialization to LEM. One particular choice is described in Section B.3 of the **SM**. We report results for the EigenWorms dataset in Table 10 of **SM** where we observe that fixing the hyperparameter $\Delta t =1$ and using chrono-initialization improved the baseline LEM (with fixed $\Delta t = 1$) results from $57.9\%$ (on average) to $88.2\%$. However, this still does not beat the results of our LEM model with a tuned $\Delta t$ ($92.3\%$). Moreover, tuning $\Delta t$ while initializing with our variant of chrono did not lead to any better results. Thus, we concluded that chrono initialization can be quite useful in the context of LEM if one wishes to fix the value of $\Delta t$. Nevertheless, further investigation is required in order to study chrono-type initializations in the context of LEM and we thank you again for pointing this out to us.

---

> > ### Comment · Reviewer_M6L8 · 2021-11-25
> > **Follow-up**
> >
> > Thanks a lot for your efforts on adding the Chrono initialization results, as well as studying its applicability to LEM. Can you say a bit more about if/how you tuned the hyperparameters for training the Chrono-LSTM baseline? Was the tuning of hyperparameters in App. A done for LEM only, or also for the Chrono-LSTM and other baselines?
> >
> > Regarding model size on psMNIST: Sure that's a valid argument, but in that case I'd like you to explicitly mention this caveat in the paper, (if accepted). This seems necessary especially because there are claims related to SOTA here (based on other papers) when it might actually mean "SOTA for 128 units" --- that too only if the hyperparameters for Chrono-LSTM baseline were reasonably well-tuned. Do you agree with these concerns?

---

> > > ### Author Response · Authors · 2021-11-26
> > > **Reply to Follow-up of Reviewer M6L8**
> > >
> > > We thank the reviewer for reading our detailed reply. Regarding your question about whether we report chrono-LSTM results after a fine-tuning procedure or not, we would like to point out that indeed, all the models that we have run have been fine-tuned with exactly the same fine-tuning protocol as LEM, mentioned in **SM** A and this very much includes the chrono-LSTM model too. In fact, we had already mentioned this in the caption of Table 3 (main text) (see also Tables 4 and 5 for other models) and apologize for not having explicitly mentioned it in Tables 1 and 2 as well. Given your suggestion, we will certainly include this fact explicitly in a camera-ready version of our paper (if accepted). Moreover, we will explicitly state all the fine-tuned hyperparameters for the models that we have run in a table in the **SM**, analogous to Table 8 for LEM, in the final version of the paper.
> > >
> > > Regarding your point about the psMNIST results, we agree with you that care must be taken in reporting results in the correct context, particularly with claims about SOTA. The phrasing about psMNIST results in the revised version of our paper reads: *"Additionally on psMNIST, LEM performs as well as coRNN, which has been SOTA among single-layer RNNs on this task"*. To the best of our knowledge, coRNN is SOTA among single-layer RNNs on this task. However, we cannot substantiate claims of other papers here. Hence, we propose to rephrase the sentence to: *"Additionally on psMNIST with 128 hidden units, LEM performs as well as the best of other models considered here."*

---

> > > > ### Comment · Reviewer_M6L8 · 2021-11-26
> > > > **Reply**
> > > >
> > > > Thank you. That sufficiently addresses my concerns related to experimental results.

---

### Official Review · Reviewer_4Tp1 · 2021-11-03

**Correctness:** 4
**Technical Novelty And Significance:** 3
**Empirical Novelty And Significance:** 3
**Recommendation:** 8
**Confidence:** 4

**Main Review:**

According to my initial review, it is a nice paper with a lot of empirical and theoretical evidence to support the claims. I have below minor concerns/comments:

One of my concerns is on the delta_t (not in bold), which is a critical hyper-parameter in discretization ODEs and turns out to be essential for performance.

-- First, the bound of proposition 4.1 could be pretty significant, which is O(sqrt(n)) given the condition that delta_t<=0.5.  With this in mind, it seems the uniform norm in proposition 4.2 could also be large unless delta_t is very small, especially for huge N. However, if delta_t is too small, the value of partial gradient (i.e., equation 7) could be pretty small.

-- Second, it seems delta_t is a key hyper-parameter for performance for different tasks. Although the authors provide some motivations for selecting delta_t, the motivation heavily depends on the understanding (experience) of the tasks and datasets. Not sure if the users have some systematic way of choosing that hyper-parameter. It is still possible that larger delta_t could still lead to exploding gradient issue, while too small delta_t might result in slow learning.

-- One conclusion that the authors have is that small delta_t would benefit tasks that require long dependency. I can imagine that small delta_t could be more friendly to mitigate the gradient exploding issue, especially for the extremely long input sequence. But there should be a sweet spot between 0 and 1 (the default value) for tasks that require long-term dependency.

My second minor concern is the experiments.

-- One suggestion is to compare with some existing ODE-based methods in more tasks, if possible.

-- The other small thought could be a bit far away from the RNN scope. Basically, for many audio and natural language processing tasks, bidirectional model, CNN, and transformer-style models are pretty common choices. I am not sure if the authors are further interested in comparing in these scenarios. For example, the bidirectional modeling (and using CNN to reduce time resolution) could help handle long-term dependency for LSTM and GRU.

Finally, I am not sure if I aligned with some claims in the draft.

-- I would not prefer that the authors claim that the proposed methods outperform other recurrent models in ASR. I would more prefer directly saying that the proposed methods work very well for keyword spotting.
For claiming ASR, the authors could consider using some widely used small/medium-size data sets (TIMIT, WSJ) and try CTC recognizers [1]. It could be interesting to see the delta_t (in bold) distribution for representations learned by phone-CTC, character-CTC, and word-CTC. It could be another piece of evidence to support the multiscale potential of the proposed method.

-- I am also not sure if it is good to claim that language modeling and speech recognition do not need long-term dependency, although there are rich short-term patterns. I would recommend the authors look into the k-word frequency pattern and delta_t (in bold) and their correlation.

[1] https://www.cs.toronto.edu/~graves/icml_2006.pdf

**Summary Of The Paper:**

The paper tries to propose a new recurrent architecture that could address the well-known issues (of recurrent models) like vanishing gradient and exploding gradient. The architecture is a kind of a realization of numerical discretization of ODEs using an implicit-explicit time-stepping scheme. The authors clearly explain why the designed model architecture can capture multiscale data. The authors also connect the proposed method to vanilla LSTM, Hodgkin-Huxley equations, and heterogeneous multiscale methods for ODEs.

The authors also try to provide theoretical evidence that the proposed methods mitigate the gradient exploding and vanishing issues (under some conditions) while learning informative representations for long/short sequence data.

The authors tried on many tasks across different domains, and the proposed methods consistently outperformed the baselines. The benefits are pretty significant in some tasks.

**Summary Of The Review:**

Currently, I think the paper is a good one for ICLR. I incline to accept it. I am also open for further discussions and adjusting my scores if needed.

---

> ### Author Response · Authors · 2021-11-19
> **Reply to Reviewer 4Tp1**
>
> We start by thanking the reviewer for your appreciation of the merits of our paper and your welcome suggestions to improve it. Below, we address the very valid concerns raised by the reviewer and thank the reviewer in advance for their patience in reading our detailed reply.
>
> * Your concern about the choice of the hyperparameter $\Delta t$ is clearly valid. The bounds (4) on the hidden states (we have increased the range to $\Delta t < 2$) and (6) on the hidden states gradients do grow polynomially (with small powers) with respect to $t_n = n \Delta t$, with $n$ denoting sequence length. By setting $\Delta t \sim \frac{1}{n}$, we can indeed obtain a uniform, with respect to sequence length $n$, bound on hidden states and their gradients. Consequently, as you pointed out, bound (7) does imply that the gradient could be small, particularly for very long sequences $n >> 1$. However, we would like to point that these upper bounds are worst-case guarantees and may be quite pessimistic for a given problem at hand. Hence, one should tune the hyperparameter $\Delta t$ by a fairly standard tuning procedure (details in **SM** Section A) and the resulting values for $\Delta t$ for all our experiments are shown in **SM** Table 8. We observe from this table that for many tasks, no tuning is necessary and a default value of $\Delta t =1$ suffices. On the other hand, the smallest value of $\Delta t$ was required for the EigenWorms dataset with a very long sequence length of $n=17984$. Based on your excellent suggestion, we further investigated if one can observe a quantitative relationship ("sweet spot") between $\Delta t$ and $n$ and realized that $\Delta t \sim \frac{1}{\sqrt{n}}$ led to  nearly optimal performance for the EigenWorms dataset (see the sensitivity study in **SM** Figure 2). Motivated by this, we set  $\Delta t = \frac{1}{\sqrt{n}}$ and trained LEM for the adding problem with sequence lengths, varying by an order of magnitude. The results presented in **SM** Figure 3 show that this choice of $\Delta t$ led to very low test errors for every sequence length. Moreover, the number of training steps to attain this low test error did not vary with respect to sequence length. This suggests that the gradient does not become too small and the model learns in a relatively small number of training steps. Based on these empirical results, we conclude that a $\Delta t$ in the range of $\frac{1}{\sqrt{n}} \leq \Delta t \leq 1$ suffices in practice and an optimal value can be found within this range by hyperparameter tuning, especially for problems with very long sequence lengths. A detailed discussion on the choice of $\Delta t$ is now added to the **SM** in the section $B.1$. Moreover, we would like to point out that the training of LEM for all the diverse tasks that we considered, required a relatively small number of training steps and did not depend on the underlying sequence length.
>
> * We would like to point out that we had already compared our model to several ODE based RNNs such as anti-symmetric RNN, Lipschitz RNN, CoRNN, Unicornn, NRDE and TARNN. We also have compared with FastRNN and FastGRNN which can be interpreted in terms of discretized ODEs. Following your suggestion, we have ensured that we report results for at least 3 ODE based RNNs in every learning task now. In particular, other ODE based RNNs are the closest competitors of LEM in several tasks with long sequence lengths. Nevertheless, LEM is able to outperform these ODE based RNNs.
>
> * Following your suggestion, we implemented a bi-directional LSTM with a one-dimensional convolutional filter on the Eigenworms dataset, which has very long sequence lengths. The result is now reported in Table 2. We see from the table that the bi-directional LSTM marginally improved the baseline LSTM result, with a reduction in the standard deviation. However, this result was significantly below accuracies of other models on this task. We fully agree with your point that adding bi-directionality and convolutional filtering could help with language modeling and speech recognition for LSTM type architectures. It would be interesting to see if adding these features would further enhance the performance of LEM too. Given the time constraints as well as page limits, we defer this interesting investigation to a follow-up paper.
>
> * We agree with your contention that speech recognition is much more general than keyword spotting. Hence, we have now replaced "speech recognition" with "keyword spotting" in all our claims about the performance of LEM. Your suggestion on using LEM for other speech recognition benchmarks such as TIMIT is very interesting. Given the constraints of time and page length, we defer a systematic study of LEM on TIMIT, including the use of CTC recognizers, to a follow-up paper.
>
> * Following your suggestion, we have now removed any possibly misleading statements that say that there are no long-term dependencies in speech recognition and language modeling.

---

### Author Response · Authors · 2021-11-19
**Reply to all the reviewers**

At the outset, we would like to thank all four reviewers for their thorough and patient reading of our article. Their fair criticism and constructive suggestions have enabled us to improve the quality of our article. A revised version of the article is uploaded. We proceed to answer the points raised by each of the reviewers individually, below.

We would also like to point out that all the references to page numbers, sections, figures, tables, equation numbers and references, refer to those in the revised version.

---

### Decision · Program_Chairs · 2022-01-20

**Decision:**

Accept (Spotlight)

**Comment:**

The paper proposes a new recurrent architecture based on discretization of ODEs which allow for learning multi-scale representations and help with the vanishing gradient problem.
The reviewers all agree this architecture is novel and provide substantial theoretical and empirical evidence.
A strong accept.